# Supralinear dendritic integration in murine dendrite-targeting interneurons

**Simonas Griesius, Amy Richardson, Dimitri Michael Kullmann***

Department of Clinical Experimental and Epilepsy, UCL Queen Square Institute of Neurology, University College London, London, United Kingdom

## eLife Assessment

In this manuscript, Griesius et al analyze the dendritic integration properties of NDNF and OLM interneurons, and suggest that the supralinear NMDA receptor-dependent synaptic integration may be associated with dendritic calcium transients only in NDNF interneurons. These findings are **important** because they suggest there might be functional heterogeneities in the mechanisms underlying synaptic integration in different classes of interneurons of the mouse neocortex and hippocampus. The revised work remains **incomplete** due to remaining concerns about experimental methodology, cell health, and lack of dendritic Na-spikes which have been recorded in previous works.

***For correspondence:**
d.kullmann@ucl.ac.uk

**Competing interest:** The authors declare that no competing interests exist.

**Abstract** Non-linear summation of synaptic inputs to the dendrites of pyramidal neurons has been proposed to increase the computation capacity of neurons through coincidence detection, signal amplification, and additional logic operations such as XOR. Supralinear dendritic integration has been documented extensively in principal neurons, mediated by several voltage-dependent conductances. It has also been reported in parvalbumin-positive hippocampal basket cells, in dendrites innervated by feedback excitatory synapses. Whether other interneurons, which support feed-forward or feedback inhibition of principal neuron dendrites, also exhibit local non-linear integration of synaptic excitation is not known. Here, we use patch-clamp electrophysiology, and two-photon calcium imaging and glutamate uncaging, to show that supralinear dendritic integration of near-synchronous spatially clustered glutamate-receptor mediated depolarization occurs in NDNF-positive neurogliaform cells and oriens-lacunosum moleculare interneurons in the mouse hippocampus. Supralinear summation was detected via recordings of somatic depolarizations elicited by uncaging of glutamate on dendritic fragments, and, in neurogliaform cells, by concurrent imaging of dendritic calcium transients. Supralinearity was abolished by blocking NMDA receptors (NMDARs) but resisted blockade of voltage-gated sodium channels. Blocking L-type calcium channels abolished supralinear calcium signalling but only had a minor effect on voltage supralinearity. Dendritic boosting of spatially clustered synaptic signals argues for previously unappreciated computational complexity in dendrite-projecting inhibitory cells of the hippocampus.

## Introduction

Accumulating evidence indicates that dendrites do more than merely summate excitatory and inhibitory synaptic inputs (*Makarov et al., 2023*; *Mikulasch et al., 2023*; *Poirazi and Papoutsi, 2020*; *Larkum et al., 2022*; *Stuart and Spruston, 2015*; *Tran-Van-Minh et al., 2015*). Dendrites endow neurons with increased computational capacity (*Lorenzo et al., 2022*) and the ability to act as multi-level hierarchical networks (*Kim et al., 2023*; *Ujfalussy et al., 2018*) or even to reproduce some features of artificial and deep neural networks (*Beniaguev et al., 2021*; *Wybo et al., 2023*). The

underlying mechanisms include extensive dendritic arborisation, compartmentalisation, synaptic plasticity and expression of specific receptors and ion channels that in turn facilitate nonlinear input summation. Nonlinear dendritic integration of excitatory inputs in particular can potentially support signal amplification (*Poleg-Polsky, 2019*), coincidence detection (*Leugering et al., 2023*), XOR logic gating (*Gidon et al., 2020*; *Magó et al., 2021*), and computing prediction errors (*Mikulasch et al., 2023*; *Wybo et al., 2023*). One striking example of non-linear integration is local supralinear summation of signals that arrive at a dendritic fragment in a narrow time window, a phenomenon that can be investigated by exploiting the spatial and temporal precision of somatic depolarizations elicited by multi-photon glutamate uncaging (*Cornford et al., 2019*; *Larkum et al., 2009*; *Katona et al., 2011*; *Tran-Van-Minh et al., 2016*; *Masala et al., 2023*; *Chiovini et al., 2014*). Several receptors and channels have been implicated in supralinear summation in principal neurons, including NMDARs (*Cornford et al., 2019*; *Larkum et al., 2009*; *Katona et al., 2011*; *Tran-Van-Minh et al., 2016*; *Chiovini et al., 2014*; *Testa-Silva et al., 2022*; *Lafourcade et al., 2022*; *Losonczy and Magee, 2006*; *Camiré et al., 2018*; *Camiré and Topolnik, 2014*; *Branco and Häusser, 2011*), calcium-permeable AMPARs (*Chiovini et al., 2014*; *Camiré et al., 2018*; *Camiré and Topolnik, 2014*), and voltage-gated sodium channels (*Masala et al., 2023*; *Chiovini et al., 2014*; *Losonczy and Magee, 2006*; *Camiré et al., 2018*; *Branco and Häusser, 2011*; *Ariav et al., 2003*), calcium channels (*Tran-Van-Minh et al., 2016*; *Chiovini et al., 2014*; *Camiré et al., 2018*; *Branco and Häusser, 2011*), and potassium channels (*Harnett et al., 2013*; *Hoffman et al., 1997*). Closely related to voltage supralinearity is the occurrence of dendritic calcium transients which can be detected with calcium-sensitive fluorescent indicators. Such calcium transients are implicated in synaptic plasticity and result not only from influx via voltage-dependent channels but also release from intracellular stores (*Camiré et al., 2018*; *Camiré and Topolnik, 2014*; *Harvey and Collingridge, 1992*; *Padamsey et al., 2019*).

Inhibitory interneurons have a crucial role in brain circuit and network excitability, rhythmicity, and computations (*Freund and Buzsáki, 1998*; *Hertäg and Sprekeler, 2020*; *Tzilivaki et al., 2023*). However, in comparison with principal cells, relatively little is known about non-linear dendritic integration in interneurons. Early evidence for NMDAR-dependent supralinear summation of EPSPs, and dendritic calcium transients, in hippocampal CA1 interneurons was not accompanied by molecular or genetic markers of interneuron type (*Katona et al., 2011*). Subsequent studies in parvalbumin-positive (PV+) interneurons reported supralinearly summating EPSPs and dendritic calcium transients (*Chiovini et al., 2014*; *Camiré et al., 2018*; *Camiré and Topolnik, 2014*). Using multiphoton glutamate uncaging for increased spatial precision, *Cornford et al., 2019* also showed evidence for NMDAR-dependent supralinear EPSP summation in PV+ basket cells, although this was only seen in dendrites receiving feedback excitation in stratum oriens and not in dendrites receiving feed-forward excitation in stratum radiatum. Failure to detect supralinear summation in stratum radiatum dendrites of PV+ interneurons is consistent with evidence from dendritic recordings that somatic action potentials attenuate rapidly as they back-propagate, and that PV+ dendrites express abundant Kv3 family potassium channels and relatively few sodium channels (*Hu and Vervaeke, 2018*). Indeed, compartmental modelling has been used to infer that synaptic inputs to PV+ interneuron dendrites summate sublinearly, and that this helps to ensure that they do not disrupt population spike synchrony during gamma oscillations (*Kriener et al., 2022*). In contrast to the evidence for both supra- and sublinear summation in PV+ interneurons (likely to occur simultaneously in different dendrites) (*Cornford et al., 2019*), it remains unknown how synaptic inputs are integrated in other forebrain interneurons. As for other brain regions, cerebellar cortex stellate cells have been shown to exhibit sublinear summation of uncaging-evoked EPSPs whilst simultaneously exhibiting supralinear summation of dendritic calcium transients (*Tran-Van-Minh et al., 2016*).

Dysfunction of interneurons generally has been associated with a variety of psychiatric and neurological pathologies, including epilepsy (*Dinocourt et al., 2003*; *Fujiwara-Tsukamoto et al., 2010*; *Magloire et al., 2019*; *Marx et al., 2013*), schizophrenia (*Belforte et al., 2010*; *Jahangir et al., 2021*; *Marín, 2012*), autism spectrum disorder (*Marín, 2012*; *Lunden et al., 2019*), and Alzheimer's disease (*Hijazi et al., 2020*; *Verret et al., 2012*), among others (*Pelkey et al., 2017*). There is emerging evidence suggesting that impairments in dendritic integration (*Masala et al., 2023*; *Mitchell et al., 2023*; *Griesius et al., 2022*; *Nelson and Bender, 2021*; *Palmer, 2014*), and specifically in interneurons (*Masala et al., 2023*), may be important contributors to pathological phenotypes in psychiatric and neurological pathologies. We therefore set out to characterize dendritic integration in two types

of interneurons innervating the distal dendrites of CA1 pyramidal neurons: PV-negative neurogliaform interneurons, which support feed-forward inhibition, and oriens-lacunosum moleculare (OLM) cells, which additionally mediate feedback inhibition. Although their cell bodies reside in different layers, and their developmental origin and intrinsic properties are strikingly different (*Pelkey et al., 2017*; *Asgarian et al., 2019*; *Booker and Wyllie, 2021*; *Sik et al., 1995*; *Chittajallu et al., 2017*; *Price et al., 2005*; *Tricoire et al., 2010*), their axons innervate the same targets, the apical tuft dendrites of pyramidal neurons in the SLM. Neurogliaform cells receive inputs from the entorhinal cortex and the nucleus reuniens (*Chittajallu et al., 2017*), with distinct plasticity rules at synapses made by these afferents (*Mercier et al., 2022*), and a subset can be identified in stratum lacunosum-moleculare (SLM) using knock-in mice expressing cre recombinase at the neuron-derived neurotrophic factor (*Ndnf*) locus (*Chittajallu et al., 2017*; *Tricoire et al., 2010*; *Mercier et al., 2022*; *Booker and Vida, 2018*). OLM cells, in contrast, have their cell bodies and dendrites in stratum oriens, are excited by axon collaterals of both CA1 and CA3 pyramidal neurons, and can be identified using mice expressing cre at the somatostatin locus (*Taniguchi et al., 2011*).

We quantified non-linear dendritic summation using two-photon glutamate uncaging at multiple locations within a small dendritic fragment, by comparing the arithmetic sum of the uncaging-evoked EPSPs evoked individually to the result of near-synchronous glutamate uncaging (*Cornford et al., 2019*). We asked if neurogliaform and OLM interneurons exhibit non-linear summation and whether this depends on NMDARs or sodium or calcium channels. In parallel, we asked if non-linear summation could also be detected in these cell types by concurrent measurement of dendritic calcium-dependent fluorescence transients.

## Results

### Supralinear dendritic integration in a representative neurogliaform interneuron

NDNF-positive (NDNF+) neurogliaform interneurons were studied in the hippocampal CA1 stratum lacunosum/moleculare in acute slices from *Ndnf^cre^* or *Ndnf^cre/cre^* mice injected with AAV2/9-mDLX-FLEX-mCherry virus (*Figure 1a*). In order to probe the spatiotemporal dynamics of dendritic integration, glutamate was locally delivered by 2-photon uncaging of bath-perfused MNI-glutamate (3 mM) using an infrared (720 nm) pulsed laser at points along dendrites, whilst recording from their somata with a patch pipette. The absence of dendritic spines prevented identification of the sites of excitatory synapses, and the uncaging locations were therefore spaced 1–2 μm apart, either side of the selected dendritic fragment (*Cornford et al., 2019*; *Figure 1b*). The locations were kept within a dendritic length of ~15 μm, consistent with the size of interneuron dendritic domains exhibiting correlated tuning of synaptic calcium transients in vivo (*Chen et al., 2013*). Glutamate uncaging was achieved with 0.5ms pulses separated by an interval of either 100.32ms (asynchronously) or 0.32ms, near-synchronously. The number of uncaging stimuli at neighbouring locations was cumulatively increased from 1 to 12 in the near-synchronous condition (*Figure 1c*). In the illustrated example, the somatic uEPSP evoked by near-synchronous uncaging was substantially greater in amplitude and duration than predicted from the arithmetic sum of the individual somatic uEPSPs (*Figure 1d*).

In parallel with measurement of uEPSPs, we measured dendritic calcium transients by monitoring the fluorescence of the calcium sensor Fluo-4, which was included in the pipette solution, using a second pulsed infrared scanning laser tuned to 810 nm (*Figure 1e and f*). The interval between glutamate uncaging pulses in the asynchronous condition was too short to allow the dendritic Fluo-4 signal to relax to baseline in between stimuli. Instead, we examined the peak amplitude of the dendritic calcium response, either in response to asynchronous uncaging at all 12 locations (*Figure 1e*) or with an increasing number of near-synchronous stimuli (*Figure 1f*). This revealed a discontinuity, where the amplitude of the dendritic calcium fluorescence transient increased abruptly with 6 or more near-synchronous uncaging locations, and exceeded the fluorescence observed with asynchronous uncaging at all 12 locations (*Figure 1e and f*).

In order to characterize non-linear voltage integration measured at the soma, the amplitude of the observed uEPSP elicited by near-synchronous uncaging at an increasing number of locations was plotted against the arithmetic sum of the individual responses. Consistent with supralinear integration in principal neurons and at oriens dendrites in PV+ interneurons, as the number of uncaging sites was

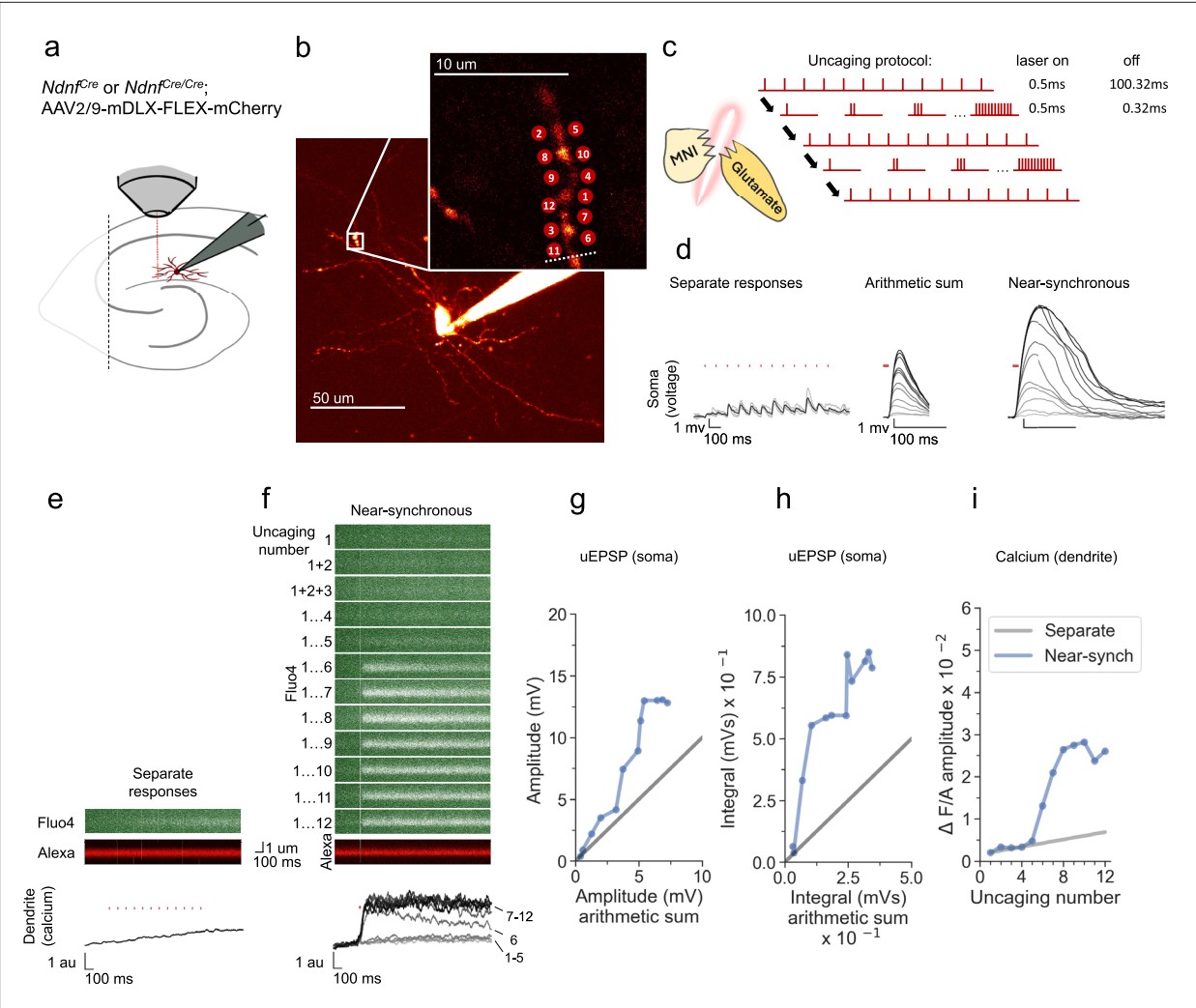

**Figure 1.** Experimental overview and representative example showing supralinear dendritic integration in a neurogliaform interneuron. (**a**) Summary depicting the mouse model, viral expression, and acute slice preparation with patch-clamp and imaging and uncaging. (**b**) Alexa-594 fluorescence Z-stack of a representative neurogliaform NDNF+interneuron, with uncaging locations and the position of the linescan used to measure dendritic calcium transients (dotted white line)(inset). The image was obtained after the experiment was concluded. (**c**) Uncaging protocol for asynchronous (separate responses) and near-synchronous conditions. (**d**) Representative somatic voltage traces across asynchronous separate response and near-synchronous conditions. Laser uncaging light stimuli are indicated by red dots above the traces. uEPSPs were elicited in the asynchronous condition with light pulses separated by an interval of 100.32ms (left; grey traces: individual sweeps; black traces average). The arithmetic sum traces were calculated by aligning and summing an increasing number of asynchronously evoked responses (middle). Near-synchronous uEPSPs were elicited with light pulses separated by an interval of 0.32ms (right). The families of traces for the arithmetic sum and near-synchronously evoked responses are shown as light to dark grey traces, indicating an increase in the number of uncaging loci from 1 to 12. (**e**) Fluo-4 and Alexa-594 linescans during asynchronous sequential uncaging at 12 loci at times indicated by the vertical red dashes. The ratio of dendritic Fluo-4 to Alexa fluorescence is shown below (au: arbitrary unit). (**f**) Linescans during near-synchronous uncaging at increasing numbers of locations (1–12, top to bottom). (**g**) Peak amplitude of the recorded near-synchronous response plotted against the amplitude of the arithmetic sum as the number of uncaging loci was increased. The line of identity is represented by a grey line. (**h**) Time integral of the recorded near-synchronous response plotted against the integral of the arithmetic sum. The line of identity is represented by a grey line. (**i**) Fluo-4 fluorescence transient amplitude, normalised by Alexa-594 fluorescence, plotted against the number of locations uncaged near-synchronously. The grey line indicates the estimated separate response amplitude interpolated from the fluorescence measured between 1 and 12 locations uncaged asynchronously (see Materials and methods).

The online version of this article includes the following figure supplement(s) for figure 1:

**Figure supplement 1.** Comparison of uEPSP time-integral and dendritic calcium in a neurogliaform cell.

increased, the recorded near-synchronously evoked uEPSPs initially summated approximately linearly, and then deviated from the line of identity upon activation of multiple uncaging locations (*Figure 1g*). The supralinearity was more marked when the uEPSP was quantified by integrating the somatic depolarization for 100ms (*Figure 1h*), as expected from the recruitment of voltage-dependent conductances. As the number of uncaging locations was increased further, the peak amplitude and integral showed a tendency to reach a plateau, consistent with a decrease in driving force as the local dendritic membrane potential approached the synaptic reversal potential.

To estimate the supralinearity of the calcium transients, we interpolated the calcium response from 1 location uncaged to the maximal fluorescence observed with all 12 locations uncaged asynchronously, and compared the peak amplitude of the calcium transients as an increasing number of locations were uncaged near-synchronously (*Figure 1i*). This revealed an abrupt deviation from the interpolated line at six locations uncaged, with a further increase as more locations were uncaged, before the observed response reached a plateau.

## Supralinear dendritic integration in neurogliaform interneurons

We examined dendritic nonlinearity in a sample of neurogliaform NDNF+ interneurons studied in the same way. This revealed robust supralinear integration of uEPSP amplitude (*Figure 2a*). The overall shape of the relationship between the observed and expected amplitude varied among cells, although its sigmoid shape persisted. We took as a measure of supralinearity in each cell the cumulative relative deviation of the observed amplitude from the expected value (see Materials and methods). Including cells that were used for subsequent blockade of NMDARs (see below), the overall average amplitude supralinearity was 58 ± 20% (average ± SEM, n=11, deviation from 0, p<0.001; *Figure 2b*). Consistent with the data from the sample neuron (*Figure 1*) the degree of voltage nonlinearity was more marked for the voltage integral measured over 100ms (integral nonlinearity 122 ± 35%), implying the involvement of slow and/or regenerating conductances (*Figure 2c and d*). As for dendritic calcium transients, six out of seven interneurons where they were measured exhibited a greater than predicted Fluo-4 fluorescence transient, with a nonlinearity estimated at 37 ± 10% (p=0.0138; *Figure 2e and f*).

Previous studies on dendritic integration have generally examined either voltage or dendritic calcium signalling in isolation, with the exception of an investigation of cerebellar layer molecular interneurons, which reported a striking dissociation, where calcium signalling exhibited supralinear summation while voltage summation was sublinear (*Tran-Van-Minh et al., 2016*), contrasting with the present results. The present findings suggest that dendritic regenerative currents are calcium-permeable, and that the calcium supralinearity is causally downstream from the voltage supralinearity. This principle was supported by the similar sigmoid shapes of the observed near-synchronous uEPSP amplitudes and calcium transients in the cell illustrated in *Figure 1*. Some cells exhibited a prominent plateau potential upon near-synchronous uncaging, with a peak calcium transient proportional to the uEPSP integral (Pearsons' correlation coefficient r=0.95; *Figure 1—figure supplement 1*). However, across the sample of 7 cells, the correlation coefficient relating calcium to voltage supralinearity varied substantially and was not systematically greater for voltage integral (0.44±0.16, average ± SEM) than for peak uEPSP amplitude (0.47±0.18).

Among possible sources of variability for voltage supralinearity, we did not observe a significant monotonic dependence on the average amplitude of individual uEPSPs, distance from the uncaging location along the dendrite to the soma, dendrite order, or dendrite diameter (*Figure 2—figure supplement 1*).

## Sublinear dendritic integration in neurogliaform interneurons with NMDARs blocked

Several conductances have been implicated in supralinear integration in principal neuron dendrites, including NMDARs (*Cornford et al., 2019*; *Larkum et al., 2009*; *Katona et al., 2011*; *Tran-Van-Minh et al., 2016*; *Chiovini et al., 2014*; *Testa-Silva et al., 2022*; *Lafourcade et al., 2022*; *Losonczy and Magee, 2006*; *Camiré et al., 2018*; *Camiré and Topolnik, 2014*; *Branco and Häusser, 2011*), sodium channels (*Masala et al., 2023*; *Chiovini et al., 2014*; *Losonczy and Magee, 2006*; *Camiré et al., 2018*; *Branco and Häusser, 2011*; *Ariav et al., 2003*), and calcium channels (*Tran-Van-Minh et al., 2016*; *Chiovini et al., 2014*; *Camiré et al., 2018*; *Branco and Häusser, 2011*). We asked whether they have similar roles in neurogliaform cells.

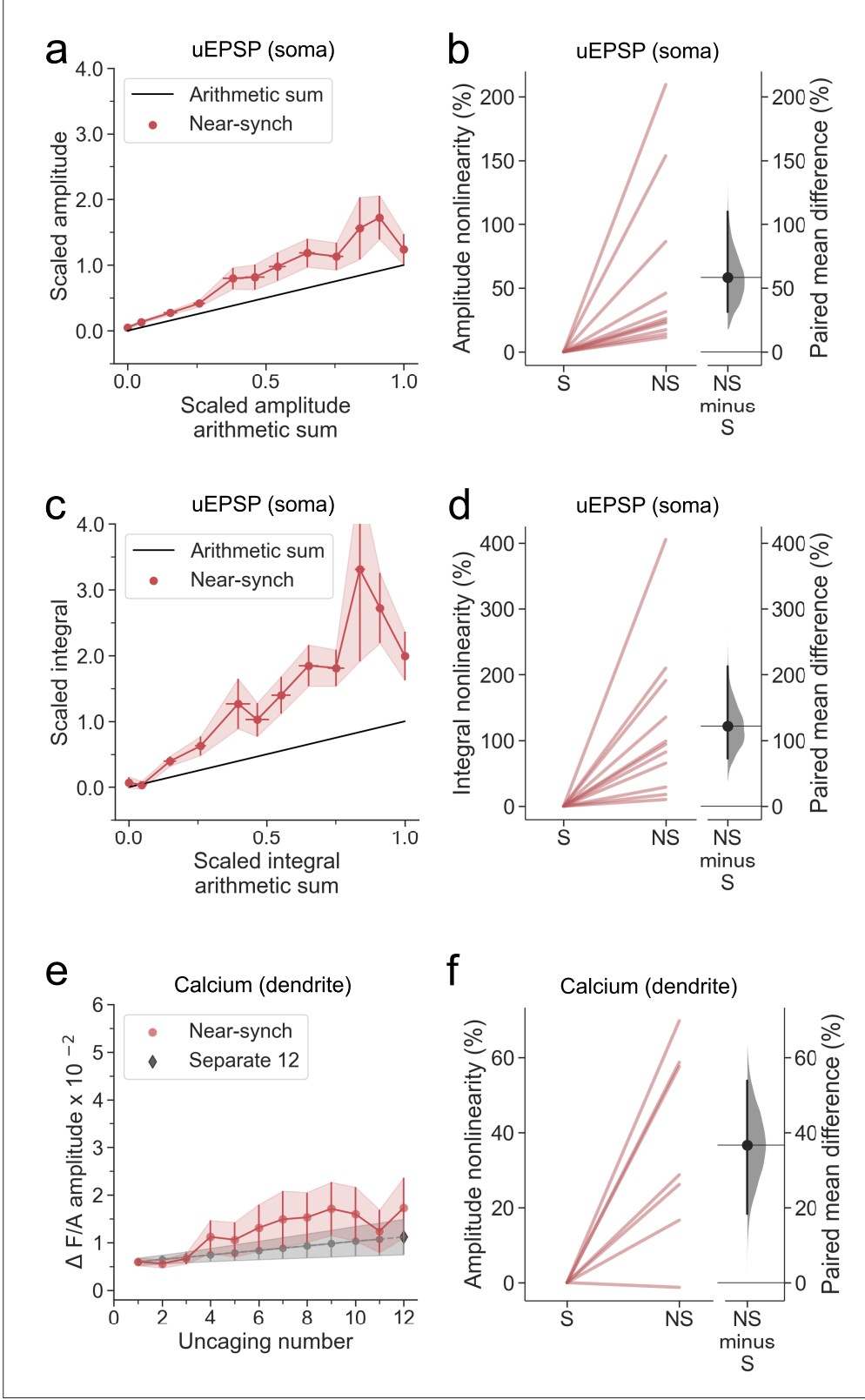

**Figure 2.** Summary of supralinear dendritic integration in neurogliaform interneurons. (**a**) Average scaled amplitude of recorded near-synchronous uEPSPs plotted against scaled amplitude of the arithmetic sum of asynchronously evoked uEPSPs as the cumulative number of uncaging locations was increased. The scaled arithmetic sum overlaps with the line of identity. Error bars: SEM (n=11 dendrites in 11 cells from 9 animals).(**b**)

*Figure 2 continued on next page*

*Figure 2 continued*

Amplitude nonlinearity quantified for data shown in (**a**). The paired mean difference between near-synchronous (NS) and asynchronous separate responses (**S**) is 56% [95%CI 27%, 108%]. p<0.001 (two-sided permutation t-test compared to 0%). (**c**) Average scaled near-synchronous uEPSP integral plotted against scaled arithmetic sum of asynchronous separate responses, for the same dataset. The scaled arithmetic sum overlaps with the line of identity. (**d**) Integral nonlinearity corresponding to panel (**c**). The paired mean difference between near-synchronous (NS) and asynchronous separate responses (**S**) is 122% [95%CI 73%, 213%]. p<0.001 (two-sided permutation t-test compared to 0%). (**e**) Average Fluo-4/Alexa-594 fluorescence transients plotted against the cumulative number of uncaging locations. The grey line indicates the interpolated values used in the nonlinearity calculation. The interpolation was performed between the calcium transient amplitude measurement at 1 uncaging location from the near-synchronous condition and the 12th response from the asynchronous (separate responses) condition (see Materials and methods). The dark grey diamond indicates the amplitude from the 12th asynchronous response. (n=7 dendrites in 7 cells from 7 animals). (**f**) Amplitude nonlinearity corresponding to panel (**c**) (dendritic calcium). The paired mean difference between near-synchronous (NS) and asynchronous separate responses (**S**) is 37% [95%CI 18%, 54%]. p=0.0138 (two-sided permutation t-test compared to 0%). For the slopegraphs and Gardner-Altman estimation plots, paired observations are connected by coloured lines. The paired mean differences between the near-synchronous (NS) and asynchronous separate response (**S**) groups are shown in Gardner-Altman estimation plots as bootstrap sampling distributions. The mean differences are depicted as black dots. The 95% confidence intervals are indicated by the ends of the vertical error bars.

The online version of this article includes the following figure supplement(s) for figure 2:

**Figure supplement 1.** Relationships among uncaging parameters and uEPSP nonlinearity (NGF).

In a subset of cells reported in *Figure 2*, we applied the NMDAR antagonist D-AP5 by bath perfusion. This robustly abolished voltage supralinearity, whether measured as the peak amplitude of uEPSPs (p<0.001, n=6; *Figure 3a, c and d*) or uEPSP integral (p<0.001; *Figure 3e and f*). Summation of somatic uEPSPs became sublinear (p<0.001, p<0.001, for peak and integral respectively), consistent with the behaviour of a passive dendrite where the local synaptic driving force and impedance diminish as an increasing number of glutamate receptors open. Dendritic calcium fluorescence transients were also profoundly reduced (*Figure 3b, g and h*).

## Supralinear dendritic integration in neurogliaform interneurons does not require voltage-dependent sodium channels

We examined the role of voltage-gated sodium channels, which have been implicated in back-propagating action potentials in principal cells (*Buzsáki and Kandel, 1998*; *Stuart and Sakmann, 1994*; *Stuart et al., 1997*), by repeating the study in the continuous presence of tetrodotoxin (TTX). In a representative cell, robust supralinear integration was seen for both uEPSPs (*Figure 4a*) and calcium fluorescence transients (*Figure 4b*). Across all dendrites tested (n=7), both the uEPSP and the dendritic calcium transient amplitudes summated supralinearity (*Figure 4c–h*), with effect sizes closely resembling those observed in the drug-free condition (*Figure 2*). The data argue against a contribution of voltage-gated sodium channels to supralinear dendritic integration in NDNF+ neurogliaform interneurons.

## L-type voltage-dependent calcium channel blockade abolishes nonlinear dendritic calcium summation

In another group of neurogliaform cells, we examined dendritic integration in the presence of the L-type voltage-dependent calcium channel blocker nimodipine. Although the population average non-linearity was not different from 0, a subset of neurons continued to show supralinear summation for both peak and integral (*Figure 5a and c–f*), contrasting with the effect of blocking NMDARs. Dendritic calcium transients were profoundly attenuated (*Figure 5b, g and h*). We tentatively conclude that L-type calcium channels have a smaller contribution to voltage supralinearity than NMDARs, but contribute substantially to dendritic calcium influx.

## SERCA inhibition abolishes nonlinear dendritic calcium summation

We next asked if release from intracellular stores also contributes to dendritic calcium transients, as has been reported in stratum oriens interneurons (*Camiré and Topolnik, 2014*). We therefore

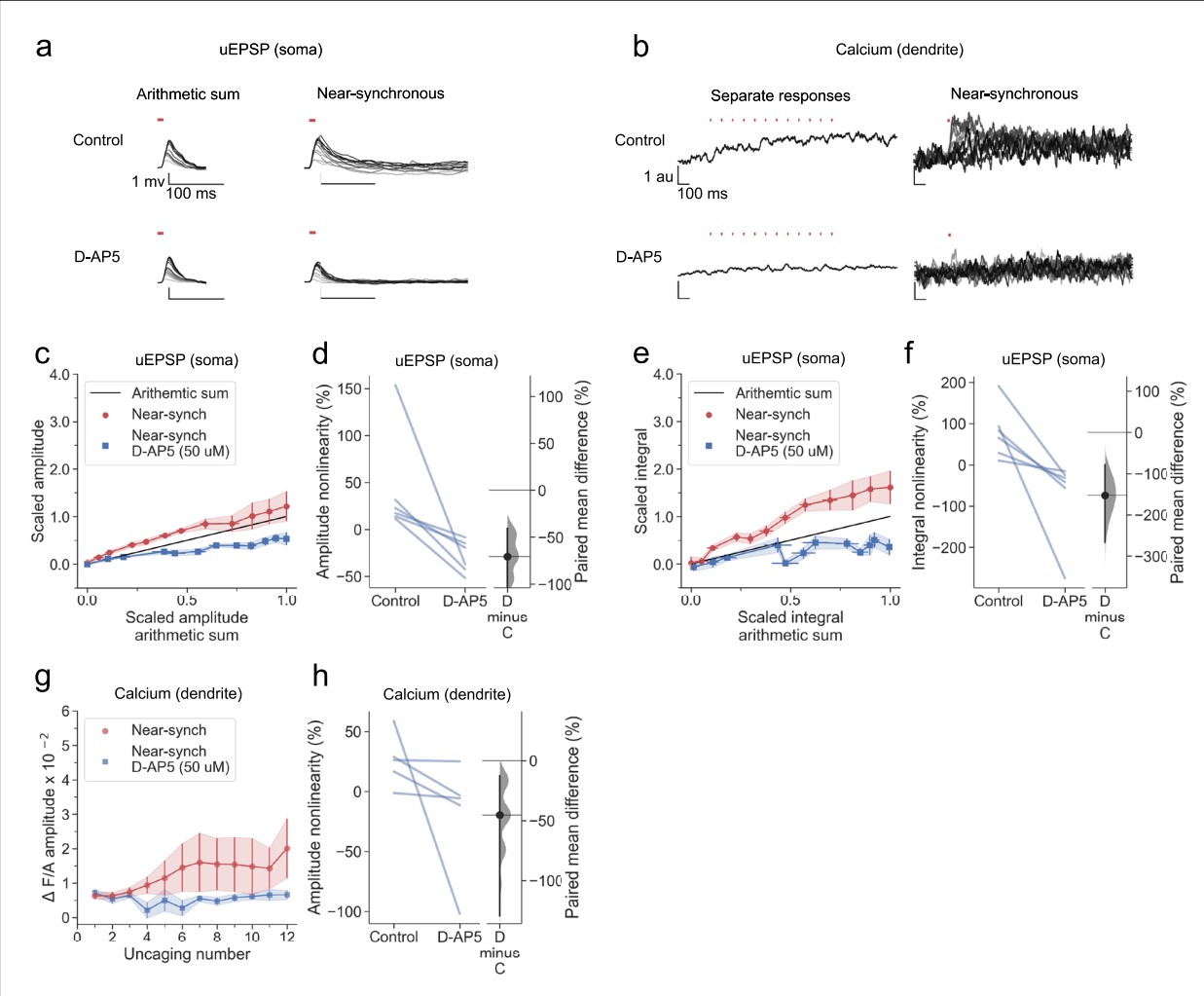

**Figure 3.** Sublinear dendritic integration in neurogliaform interneurons with NMDARs blocked. (**a**) Somatic uEPSPs evoked by uncaging glutamate in a representative neuron before (top) and after (bottom) D-AP5 (50 μM) perfusion. Uncaging protocol and data representation as in *Figures 1–2*. (**b**) Dendritic Fluo-4 traces in the same neuron, in response to asynchronous and near-synchronous uncaging, before (left) and after NMDAR blockade (right). (**c**) Scaled amplitude of the recorded near-synchronous uEPSP plotted against scaled amplitude of the arithmetic sum, as the number of uncaging locations was increased. The scaled arithmetic sum overlaps with the line of identity. Error bars: SEM (n=6 dendrites in 6 cells from 6 animals). The supralinear relationship became sublinear with NMDARs blocked. (**d**) Amplitude nonlinearity for individual neurons shown in (**c**) before and after D-AP5 perfusion. The paired mean difference between the two conditions is –71% [95% CI –147%, –41%]. p<0.001 (two-sided permutation t-test). (**e**) Scaled time-integral of uEPSPs plotted against scaled integral arithmetic sum of asynchronous separate responses, before and after NMDAR blockade. The scaled arithmetic sum overlaps with the line of identity. (**f**) Integral nonlinearity corresponding to panel (**e**). The paired mean difference between baseline and D-AP5 conditions is –153% [95% CI –267%, –78%]. p<0.001 (two-sided permutation t-test). (**g**) Fluo-4 fluorescence transients (normalised to Alexa-594) plotted against the cumulative number of uncaging locations, before and after NMDAR blockade. Data plotted as in *Figure 1*. (**h**) Amplitude nonlinearity corresponding to panel (**g**). The paired mean difference between baseline and D-AP5 conditions is –45% [95% CI –129%, –13%]. p<0.001 (two-sided permutation t-test). Slopegraphs and Gardner-Altman estimation plots as in *Figure 2*.

examined the effect of depleting calcium stores with the SERCA inhibitor cyclopiazonic acid (CPA). Although voltage supralinearity persisted (*Figure 6a and c–f*), dendritic calcium transients were abolished (*Figure 6b, g and h*).

A parsimonious explanation for the dissociation of voltage supralinearity and calcium signalling is that NMDARs are the main driver of regenerative currents in the dendrites of neurogliaform cells, and that calcium influx via NMDARs and L-type calcium channels is amplified by calcium release from intracellular stores. Overall, the robust NMDAR-mediated supralinearity demonstrated in neurogliaform cells stands in contrast to the more subtle supralinearity observed in PV+ interneurons in response to

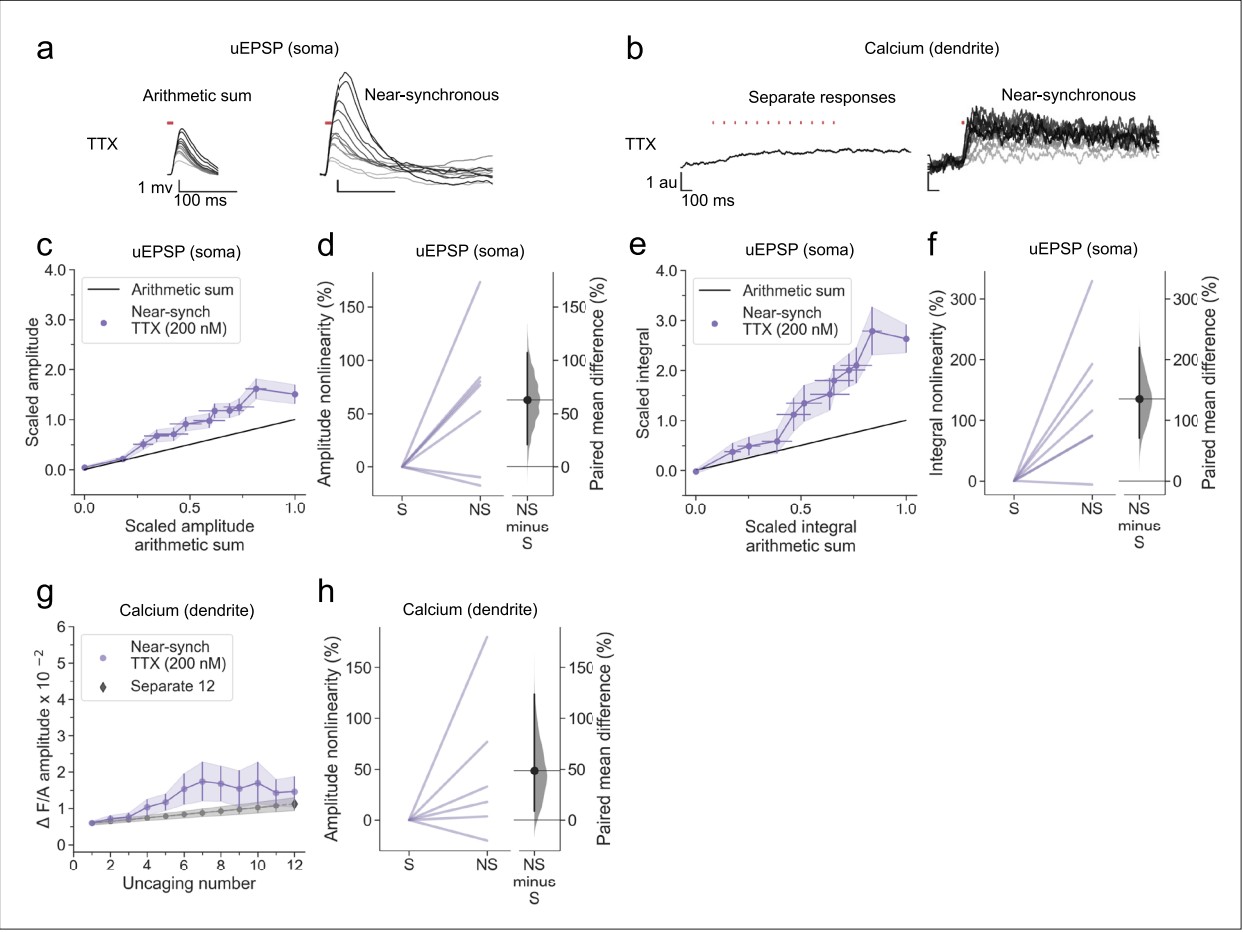

**Figure 4.** Supralinear dendritic integration in neurogliaform interneurons does not require voltage-dependent sodium channels. (**a**) Somatic uEPSPs evoked by uncaging glutamate in a representative neuron in the presence of TTX (200 nM), showing supralinear summation. (**b**) Dendritic Fluo-4 fluorescence responses in the same neuron, showing supralinear calcium signalling. (**c**) Average scaled amplitude of recorded near-synchronous uEPSP plotted against scaled amplitude arithmetic sum with increasing numbers of cumulative uncaging locations. The scaled arithmetic sum overlaps with the line of identity. Error bars: SEM (n=7 dendrites in 5 cells from 5 animals). (**d**) Amplitude nonlinearity corresponding to panel (**c**). The paired mean difference between near-synchronous (NS) and asynchronous separate responses (**S**) is 63% [95%CI 21%, 107%]. p=0.0382 (two-sided permutation t-test compared to 0%). (**e**) uEPSP amplitude nonlinearity quantified by integral, plotted in the same way as (**c**). (**f**) Integral nonlinearity corresponding to (**e**). The paired mean difference between near-synchronous (NS) and asynchronous separate responses (**S**) is 135% [95%CI 71%, 220%]. p=0.011 (two-sided permutation t-test compared to 0%). (**g**) Fluo-4 fluorescence transient amplitude (normalised to Alexa-594) plotted against the cumulative number of uncaging locations. Data plotted as in *Figure 1*. (**h**) Amplitude nonlinearity corresponding to panel (**g**). The paired mean difference between near-synchronous (NS) and asynchronous separate responses (**S**) is 48% [95%CI 9%, 123%]. p=0.088 (two-sided permutation t-test compared to 0%). Slopegraphs and Gardner-Altman estimation plots as in *Figure 2*.

an identical glutamate uncaging protocol, which was confined to dendrites in stratum oriens (*Cornford et al., 2019*).

## Supralinear dendritic integration in OLM interneurons

Apical tuft dendrites of pyramidal neurons are innervated not only by neurogliaform cells but also by OLM interneurons, which exhibit numerous differences with respect to their developmental origin, morphology and intrinsic and synaptic properties. We asked whether the robust supralinear summation and calcium signalling exhibited by neurogliaform cells are also a feature of OLM cells, by repeating the experiments using multiphoton uncaging on dendrites in stratum oriens. OLM cells were identified in acute hippocampal slices from *Sst^{Cre/Cre}* x *Rosa^{LSL-tdTomato}* mice, with dendrites running parallel to stratum pyramidale and an axon, where visualised, extending through stratum pyramidale and stratum radiatum (*Figure 7a and b*).

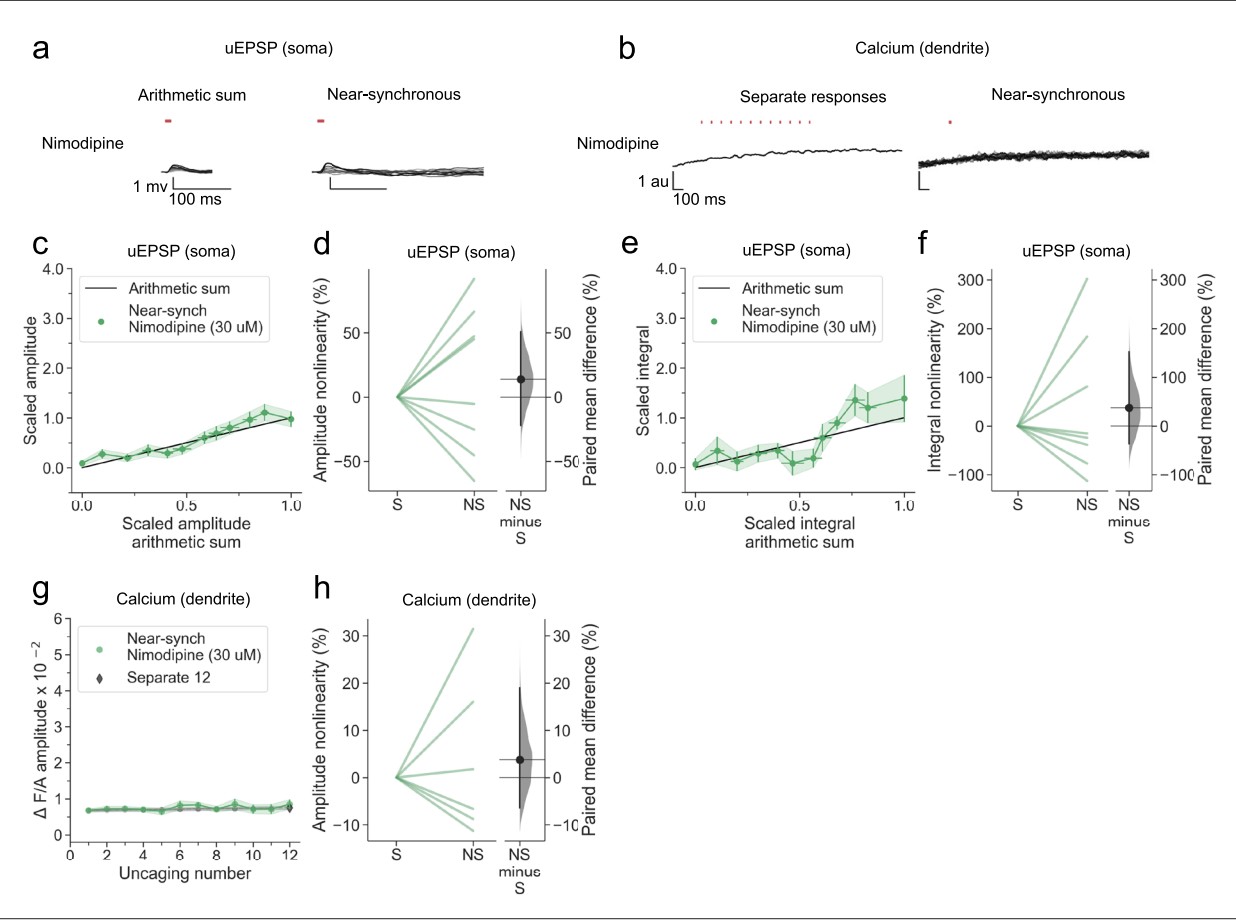

**Figure 5.** L-type voltage-dependent calcium channel blockade abolishes nonlinear dendritic calcium summation. (**a**) Somatic uEPSPs evoked by glutamate uncaging in one neuron in the presence of nimodipine (30 μM), showing supralinear uEPSP integration. (**b**) Dendritic Fluo-4 fluorescence transients in the same neuron, showing absence of supralinear integration. (**c**) Average scaled amplitude of the recorded near-synchronous uEPSP plotted against the arithmetic sum of the scaled asynchronous uEPSPs, as the cumulative number of uncaging locations was increased. The scaled arithmetic sum overlaps with the line of identity. Error bars:SEM (n=8 dendrites in 5 cells from 3 animals). (**d**) Amplitude nonlinearity corresponding to (**c**). The paired mean difference between near-synchronous (**S**) and asynchronous separate responses (**S**) is 14% [95%CI –22%, 51%]. p=0.505 (two-sided permutation t-test compared to 0%). (**e**) uEPSP amplitude nonlinearity quantified by integral, plotted in the same way as (**c**). (**f**) Integral nonlinearity corresponding to (**e**). The paired mean difference between near-synchronous (**NS**) and asynchronous separate responses (**S**) is 37% [95%CI –37%, 153%]. p=0.497 (two-sided permutation t-test compared to 0%). (**g**) Fluo-4 fluorescence transient amplitude (normalised to Alexa-594) plotted against cumulative number of uncaging locations. Data plotted as in **Figure 1**. (**h**) Amplitude nonlinearity corresponding to (**g**). The paired mean difference between near-synchronous (**NS**) and asynchronous separate responses (**S**) is 4% [95%CI –6%, 19%]. p=0.561 (two-sided permutation t-test compared to 0%). Slopegraphs and Gardner-Altman estimation plots as in **Figure 2**.

Uncaging-evoked uEPSPs summated supralinearly in OLM cells, as detected by comparing either the peak amplitude or voltage integral of near-synchronously evoked responses to the arithmetic sum of the asynchronously evoked responses (**Figure 7c–g**).

## Linear dendritic integration in OLM interneurons with NMDARs blocked

As for neurogliaform cells, there was no obvious dependence of the magnitude of supralinearity on the average size of the individual uEPSPs, dendritic distance, dendrite order, or dendrite diameter (**Figure 7—figure supplement 1**). NMDAR blockade abolished supralinearity (**Figure 8a–e**), further underlining the similarity with neurogliaform cells. However, sublinear summation was not observed in D-AP5. Another striking difference was that the uEPSPs evoked in OLM cells (with NMDARs intact) were sub-threshold for detectable dendritic calcium signals under our recording conditions.

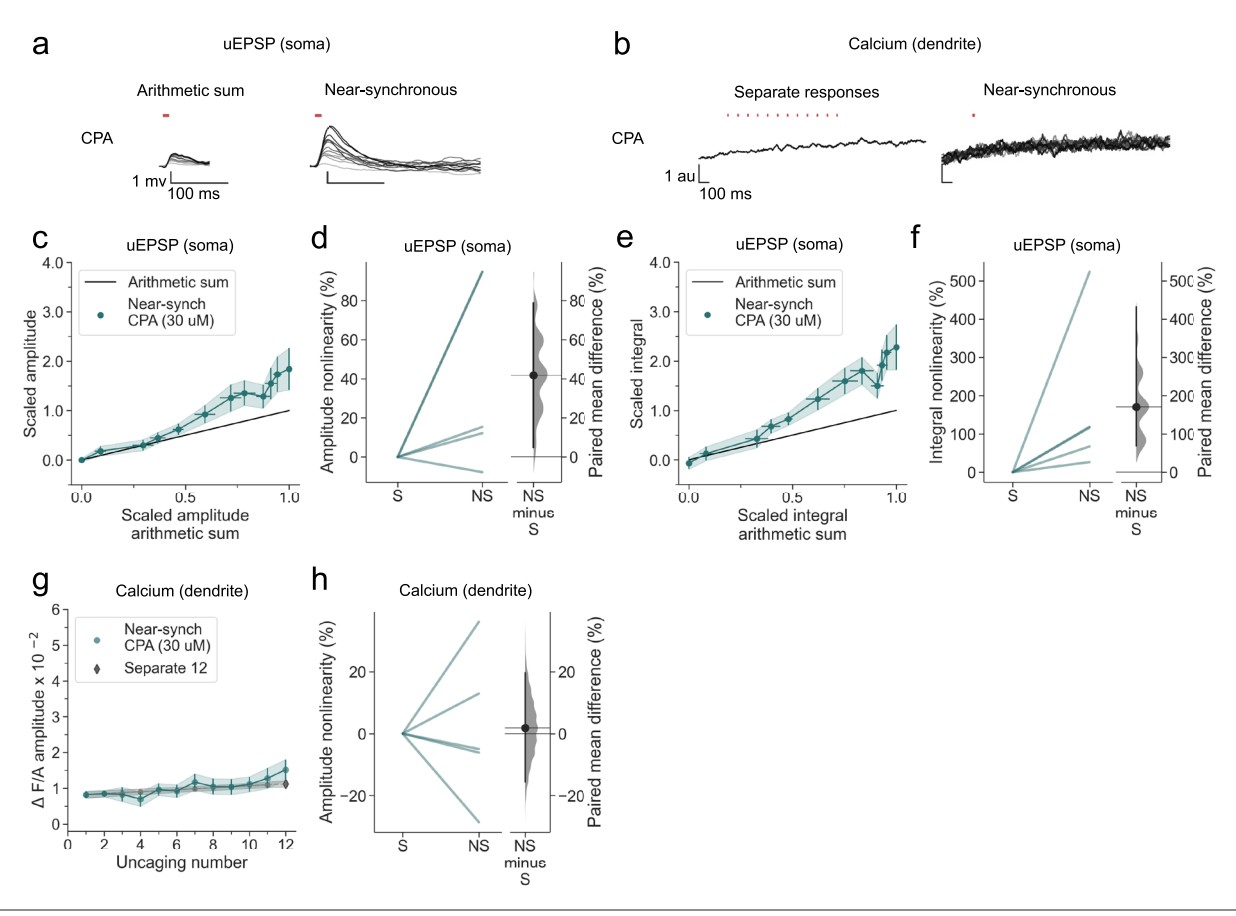

**Figure 6.** SERCA inhibition abolishes nonlinear dendritic calcium summation. (**a**) Somatic uEPSPs evoked by glutamate uncaging in one neuron in the presence of CPA (30 μM), showing supralinear uEPSP integration. (**b**) Dendritic Fluo-4 fluorescence transients in the same neuron, showing absence of supralinear integration. (**c**) Average scaled amplitude of the recorded near-synchronous uEPSP plotted against the arithmetic sum of the scaled asynchronous uEPSPs, as the cumulative number of uncaging locations was increased. The scaled arithmetic sum overlaps with the line of identity. Error bars:SEM (n=5 dendrites in 2 cells from 2 animals). (**d**) Amplitude nonlinearity corresponding to (**c**). The paired mean difference between near-synchronous (NS) and asynchronous separate responses (**S**) is 42% [95%CI 5%, 79%]. p=0.067 (two-sided permutation t-test compared to 0%). (**e**) uEPSP amplitude nonlinearity quantified by integral, plotted in the same way as (**c**). (**f**) Integral nonlinearity corresponding to (**e**). The paired mean difference between near-synchronous (NS) and asynchronous separate responses (**S**) is 171% [95%CI 69%, 432%]. p<0.001 (two-sided permutation t-test compared to 0%). (**g**) Fluo-4 fluorescence transient amplitude (normalised to Alexa-594) plotted against cumulative number of uncaging locations. Data plotted as in **Figure 1**. (**h**) Amplitude nonlinearity corresponding to (**g**). The paired mean difference between near-synchronous (NS) and asynchronous separate responses (**S**) is 2% [95%CI –16%, 20%]. p=0.754 (two-sided permutation t-test compared to 0%). Slopegraphs and Gardner-Altman estimation plots as in **Figure 2**.

## No effect of voltage-dependent sodium channel nor of L-type voltage-dependent calcium channel blockade in OLM interneurons

Neither sodium channel blockade (**Figure 8f–j**), nor calcium channel blockade (**Figure 8k–o**) prevented supralinear summation of uEPSPs in OLM interneurons. Depletion of intracellular calcium stores with CPA also had no effect (**Figure 8—figure supplement 1**). The magnitude of supralinearity in the presence of either TTX or nimodipine was no different from the magnitude in the absence of blockers. The data thus point to a model where supralinearity is mediated exclusively by NMDARs, and occurs with uEPSPs that are subthreshold for dendritic calcium signalling.

The absence of sublinear summation in D-AP5 is reminiscent of the pattern observed in stratum oriens dendrites of PV+ interneurons (**Cornford et al., 2019**), as is the relative magnitude of supra-linearity. When neurogliaform and OLM data obtained in TTX were grouped together with the data without blockers, the supralinearity for voltage integral in OLM interneurons was significantly smaller than that in neurogliaform cells (**Figure 8p**). Among possible explanations is that the local dendritic

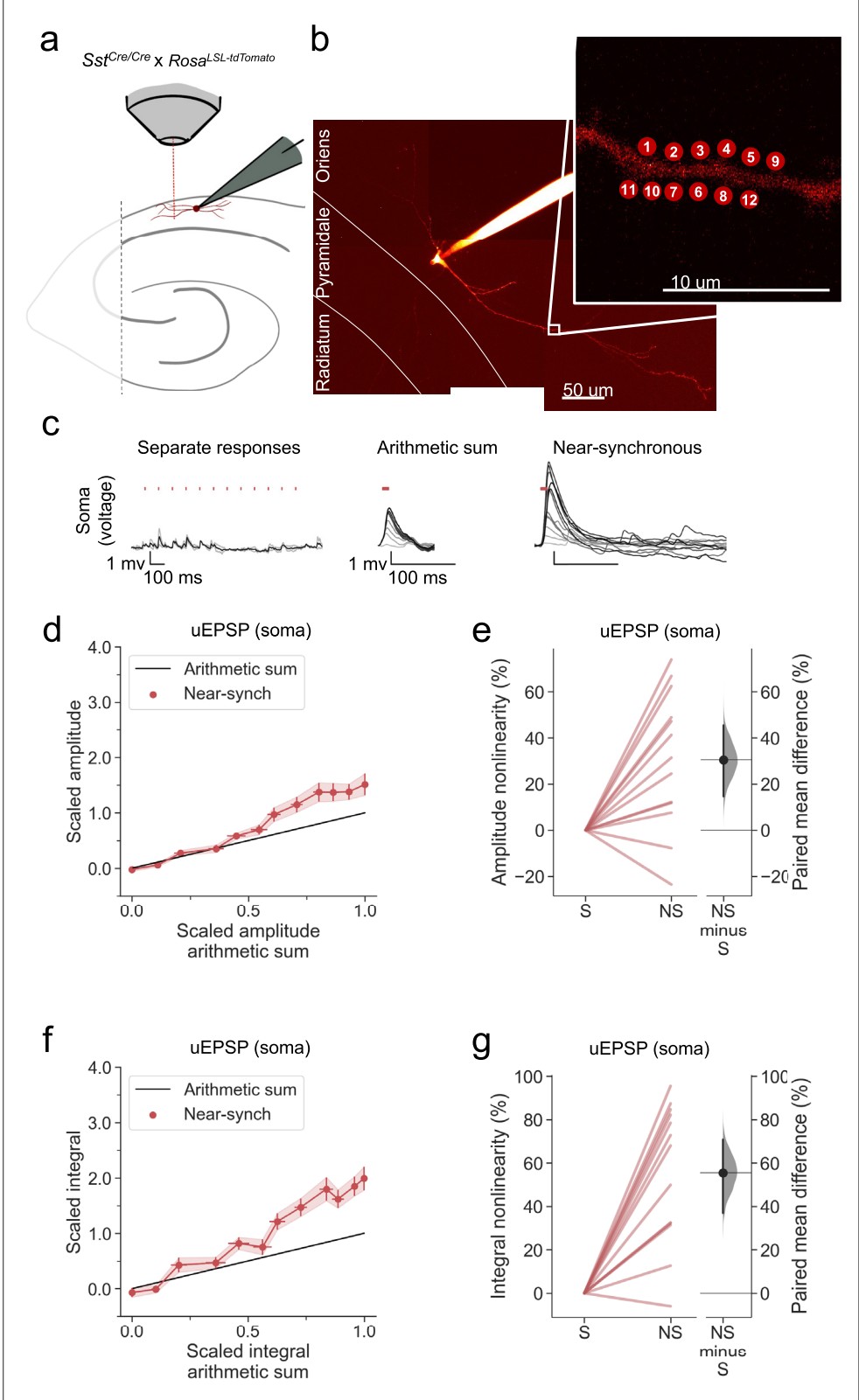

**Figure 7.** Supralinear dendritic integration in OLM interneurons. (**a**) Summary depicting the mouse model and acute slice preparation with patch-clamp and imaging and uncaging. Uncaging protocol and data representation as in *Figures 1–2* (**b**) Alexa-594 fluorescence Z-stack of a representative OLM interneuron with uncaging locations (inset). The image was obtained after the experiment was concluded. (**c**) Representative somatic voltage traces

*Figure 7 continued*

across asynchronous separate response and near-synchronous conditions, depicted as in *Figure 1*. (**d**) Average scaled amplitude of the recorded near-synchronous uEPSP plotted against the arithmetic sum of the scaled asynchronous uEPSPs, as the cumulative number of uncaging locations was increased. The scaled arithmetic sum overlaps with the line of identity. Error bars: SEM (n=13 dendrites in 13 cells from 13 animals). (**e**) Amplitude nonlinearity corresponding to (**d**). The paired mean difference between near-synchronous (NS) and asynchronous separate responses (**S**) is 31% [95%CI 15%, 46%]. *P*=0.004 (two-sided permutation t-test compared to 0%). (**f**) uEPSP amplitude nonlinearity quantified by integral, plotted in the same way as (**d**). (**g**) Integral nonlinearity corresponding to (**f**). The paired mean difference between near-synchronous (NS) and asynchronous separate responses (**S**) is 56% [95%CI 31%, 71%]. p<0.001 (two-sided permutation t-test compared to 0%). Slopegraphs and Gardner-Altman estimation plots as in *Figure 2*.

The online version of this article includes the following figure supplement(s) for figure 7:

**Figure supplement 1.** Relationships among uncaging parameters and uEPSP nonlinearity (OLM).

---

impedance is greater in neurogliaform cells, lowering the threshold for recruitment of regenerative currents. Consistent with such a pattern, the local dendrite diameter of neurogliaform interneurons was significantly lower than that of OLM cells (*Figure 8q*). A morphological difference may thus contribute to the smaller voltage and calcium supralinearities in OLM than neurogliaform cells.

## Discussion

The main results of the present study are that both hippocampal neurogliaform interneurons and OLM cells exhibit robust NMDAR-dependent supralinear integration of dendritic uEPSPs recorded at the soma. Supralinear NMDAR-dependent integration was also observed by measuring dendritic calcium fluorescence transients in neurogliaform cells, which were sensitive to blockade of L-type calcium channels or depletion of intracellular stores. However, near-synchronous glutamate uncaging that was sufficient to elicit NMDAR-dependent boosting of uEPSPs failed to recruit detectable dendritic calcium transients in OLM cells. In contrast to principal neurons, we observed no role for voltage-gated sodium channels in dendritic integration of uEPSPs in either cell type.

Taken together, the data are consistent with a model where clustered synaptic receptor activation leads to relief of voltage-dependent block of dendritic NMDARs by magnesium ions, accounting for supralinear summation of uEPSPs recorded at the soma. In neurogliaform cells (although apparently not in OLM cells) L-type voltage-gated calcium channels are also recruited by the local depolarization, and together with calcium influx via NMDARs, the resulting local elevation of intracellular calcium triggers further release from intracellular stores, thereby amplifying the dendritic calcium concentration transient. This account of the signalling cascade underlying local calcium signalling does not exclude additional roles for metabotropic glutamate receptors and calcium influx via rectifying calcium-permeable glutamate receptors, which have been implicated in dendritic calcium transients in oriens interneurons (*Camiré and Topolnik, 2014*).

The dependence of supralinear dendritic summation on NMDARs but not voltage-gated sodium channels is reminiscent to the pattern observed at dendrites of parvalbumin-positive interneurons in stratum oriens (*Cornford et al., 2019*). NMDARs have previously been implicated in synaptic signalling in neurogliaform cells (*Chittajallu et al., 2017*; *Price et al., 2005*; *Armstrong et al., 2011*) and also contribute to long-term potentiation (*Mercier et al., 2022*). Indeed, the ratio of NMDAR- to AMPAR-mediated signalling at glutamatergic synapses on neurogliaform cells has been shown to be higher than in other hippocampal interneuron subtypes (*Chittajallu et al., 2017*). This feature may explain why NMDAR-mediated supralinear integration is especially prominent in neurogliaform cells, although a further contribution may come from their unusually thin and short dendrites, which would be expected to represent a high impedance, thus facilitating their depolarization by glutamate receptor activation.

OLM cells exhibited several differences in comparison with neurogliaform cells. First, supralinear voltage integration was smaller. Second, blockade of NMDARs led to linear, as opposed to sublinear summation of uEPSPs. And third, uEPSPs evoked by glutamate uncaging, whether asynchronous or near-synchronous, were subthreshold for robust dendritic calcium transients. These differences can be explained by a relatively lower expression of NMDARs and/or lower dendritic impedance as expected

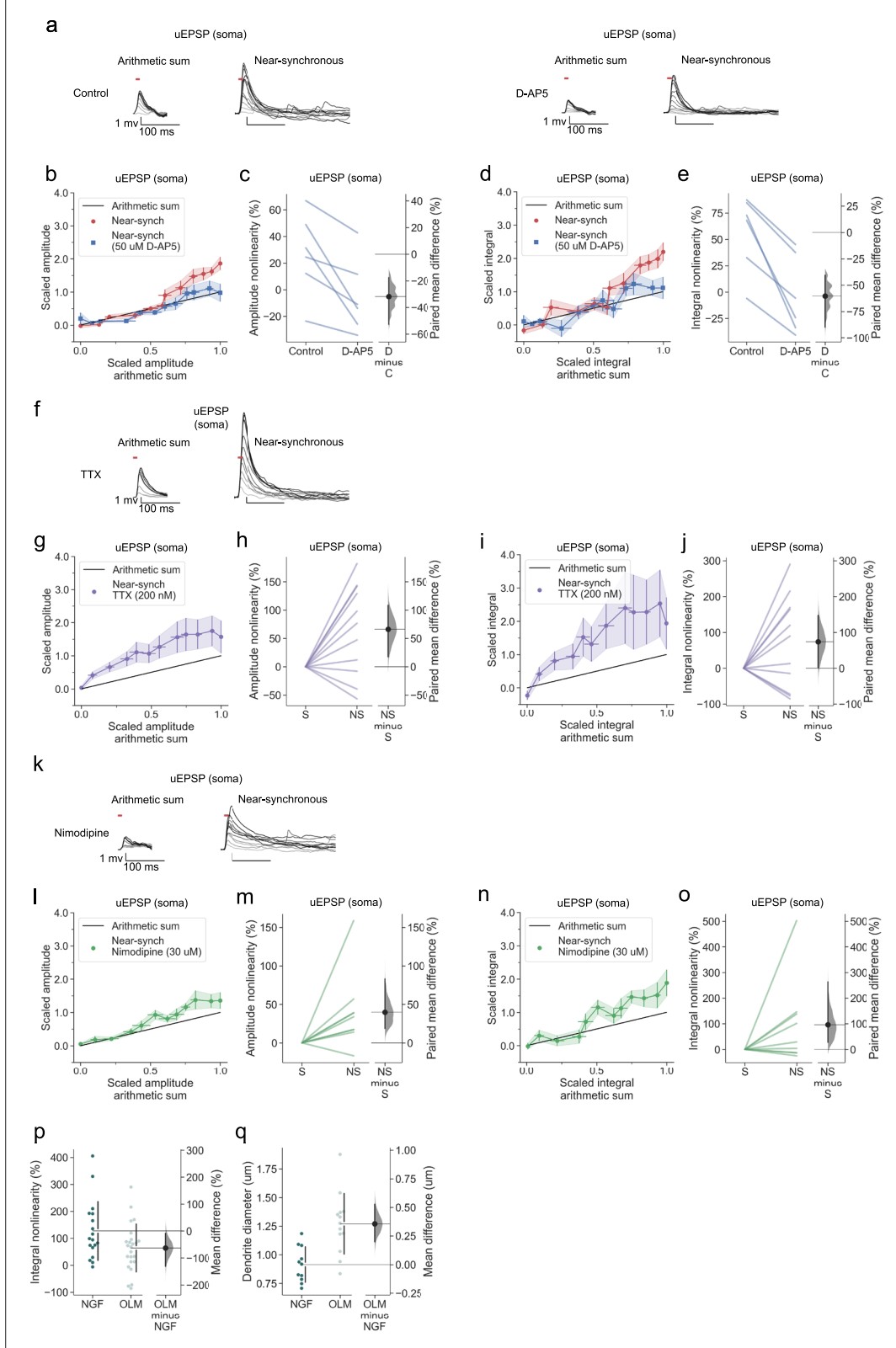

**Figure 8.** Linear dendritic integration in OLM interneurons with NMDARs blocked and no effect of voltage-dependent sodium channel nor of L-type voltage-dependent calcium channel blockade. (**a**) Somatic uEPSPs evoked by uncaging glutamate in a representative neuron before (top) and after (bottom) D-AP5 (50 μM) perfusion. Uncaging protocol and data representation as in *Figures 1–2*. (**b**) Scaled amplitude of the recorded

*Figure 8 continued on next page*

*Figure 8 continued*

near-synchronous uEPSP plotted against scaled amplitude of the arithmetic sum, as the number of uncaging locations was increased. The scaled arithmetic sum overlaps with the line of identity. Error bars: SEM (n=6 dendrites in 6 cells from 6 animals). (**c**) Amplitude nonlinearity for individual neurons shown in (**b**) before and after D-AP5 perfusion. The paired mean difference between the two conditions is –32% [95% CI –53%, –18%]. p<0.001 (two-sided permutation t-test). (**d**) Scaled time-integral of uEPSPs plotted against scaled integral arithmetic sum of asynchronous separate responses, before and after NMDAR blockade. (**e**) Integral nonlinearity corresponding to panel (**d**). The paired mean difference between baseline and D-AP5 conditions is –60% [95% CI –90%,–40%]. p<0.001 (two-sided permutation t-test). (**f**) Somatic uEPSPs evoked by glutamate uncaging in one neuron in the presence of TTX (200 nM), showing supralinear uEPSP integration. (**g**) Average scaled amplitude of the recorded near-synchronous uEPSP plotted against the arithmetic sum of the scaled asynchronous uEPSPs, as the cumulative number of uncaging locations was increased. Error bars: SEM (n=11 dendrites in 7 cells from 6 animals). (**h**) Amplitude nonlinearity corresponding to (**g**). The paired mean difference between near-synchronous (NS) and asynchronous separate responses (**S**) is 66% [95%CI 17%, 102%]. p=0.029 (two-sided permutation t-test compared to 0%). (**i**) uEPSP amplitude nonlinearity quantified by integral, plotted in the same way as (**g**). (**j**) Integral nonlinearity corresponding to (**i**). The paired mean difference between near-synchronous (NS) and asynchronous separate responses (**S**) is 74% [95%CI 1%, 148%]. p=0.089 (two-sided permutation t-test compared to 0%). (**k**) Somatic uEPSPs evoked by glutamate uncaging in one neuron in the presence of nimodipine (30 µM), showing supralinear uEPSP integration. (**l**) uEPSP amplitude nonlinearity, plotted in the same way as (**g**). (n=8 dendrites in 7 cells from 7 animals). (**m**) Amplitude nonlinearity corresponding to (**l**). The paired mean difference between near-synchronous (NS) and asynchronous separate responses (**S**) is 40% [95%CI 19%, 83%]. p=0.011 (two-sided permutation t-test compared to 0%). (**n**) uEPSP amplitude nonlinearity quantified by integral, plotted in the same way as (**g**). (**o**) Integral nonlinearity corresponding to (**n**). The paired mean difference between near-synchronous (NS) and asynchronous separate responses (**S**) is 97% [95%CI 27%, 263%]. p=0.066 (two-sided permutation t-test compared to 0%). (**p**) uEPSP amplitude nonlinearity quantified by integral in neurogliaform (NGF) and OLM interneurons (control and TTX experiments pooled). The mean difference between NGF and OLM integral nonlinearity is –63% [95% CI –132%, –8%]. p=0.0424 (two-sided permutation t-test). (**q**) Dendritic diameter in NGF and OLM interneurons. The mean difference between NGF and OLM 0.36 [95%CI 0.20, 0.53]. p<0.001 (two-sided permutation t-test). Slopegraphs and Gardner-Altman estimation plots as in *Figure 2*.

The online version of this article includes the following figure supplement(s) for figure 8:

**Figure supplement 1.** SERCA inhibition does not affect supralinear dendritic summation in OLM interneurons.

from larger dendritic diameter and length. Indeed, we confirmed that the dendritic diameter at the uncaging locations was larger in OLM than neurogliaform cells. A further morphological difference is that the dendrites of OLM cells are equipped with sparse thin spines (*Martina et al., 2000*; *Maccaferri et al., 2000*), although we did not visualize them. The density of both spines and NMDA receptors have also been reported to differ with distance from the soma in the dendrites of OLM cells (*Gramuntell et al., 2021*; *Guirado et al., 2014*). Whilst we saw no evidence suggesting sodium-channel-dependent supralinear voltage integration in OLM cells, OLM cells have been reported to boost dendritic signals via active sodium conductances when increasing amounts of current were injected into the dendrites (*Martina et al., 2000*). This difference may be due to the uncaging stimulation protocol remaining subthreshold for the initiation of sodium-channel-dependent boosting, as current injection seemed to remain linear until EPSPs became greater than ~20 mV in amplitude (*Martina et al., 2000*).

The present study highlights the ability of both neurogliaform and OLM neurons to perform local computations upstream of the site of axonal action potential initiation. The adaptive significance of this phenomenon remains to be determined. In principal neurons, synaptic inputs converging to common dendritic domains have been shown to exhibit correlated activity (*Iacaruso et al., 2017*; *Takahashi et al., 2012*; *Wilson et al., 2016*). Synaptically evoked calcium transients in dendritic domains of interneurons of the visual cortex have also been reported to exhibit orientation tuning that is not apparent in the global activity recorded at the soma (*Chen et al., 2013*). The dendritic nonlinearities uncovered by the present study, together with different forms of synaptic plasticity described in interneurons (*Kullmann et al., 2012*), are a candidate mechanism to achieve such an organization of information processing.

# Materials and methods

## Animals and husbandry

Adult male and female mice of varying ages were throughout the study (ages 1–3 months). Transgenic mouse lines were maintained as heterozygotes. $Ndnf^{cre/cre}$ or $Ndnf^{cre/+}$ mice (The Jackson Laboratory B6.Cg-$Ndnf^{tm1.1(folA/cre)Hze}$/J; Stock No: 028536; Bar Harbor, ME, USA), bred on a C57BL/6 background were used to target neurogliaform interneurons (**Tasic et al., 2016**). $Sst^{cre/cre}$ (The Jackson Laboratory $Sst^{tm2.1(cre)Zjh}$/J; Stock No: 013044; Bar Harbor, ME, USA) crossed with $Rosa^{LSL-tdTomato}$ mice (The Jackson Laboratory B6.Cg-$^{Gt(ROSA)26Sortm9(CAG-tdTomato)Hze}$/J; Stock No: 007909; Bar Harbor, ME, USA) were used to target OLM interneurons (**Taniguchi et al., 2011**; **Madisen et al., 2010**). Animals were group-housed under a normal 12 hr light/dark cycle. Cages were enriched with paper bedding, cardboard tube, plastic tube, and wooden chewing blocks. Mice had had unlimited access to standard laboratory chow and water. Research was conducted in accordance with the Animal (Scientific Procedures) Act 1986 Amendment Regulations (SI 2012/3039) under PPL: PP3944290.

## Surgery for viral injection

Mice of at least 6 weeks of age were anaesthetised with isoflurane and placed into a stereotaxic frame, onto a heating pad to maintain body temperature. Mice were given Metacam (0.1 mg/kg) and buprenorphine (0.02 mg/kg) subcutaneously. Bilateral craniotomies were performed, positioned 3.1 mm caudal and ±3.12 mm lateral of Bregma. Virus (AAV2/9-mDlx-FLEX-mCherry; titre >$10^{12}$ viral genomes/ml; VectorBuilder [Chicago, IL, USA]) was injected into the ventral CA1 region of both hippocampi using a Hamilton syringe 2.5 mm deep from the pia. 150 nL of virus was injected at each site at a rate of 100 nL/min. The needle was left in place for 5 min following injections before withdrawal. Mice were given 0.5 mL saline subcutaneously post-operatively to help with recovery and were monitored for 5 days following the procedure. Mice were sacrificed for experiments after a minimum of 3 weeks post-surgery.

## Brain slice preparation

Mice were sacrificed, the brains removed, and hippocampi dissected and sliced in ice-cold sucrose-based solution containing (in mM): 205 Sucrose, 10 Glucose, 26 NaHCO3, 2.5 KCl, 1.25 NaH2PO4, 0.5 CaCl2, 5 MgSO4, saturated with 95% O2 and 5% CO2. Transverse 400 µm hippocampal slices were cut using a Leica VT1200S vibrating microtome. Slices were incubated in artificial cerebrospinal fluid (aCSF) at 35 °C for 30 min and then at room temperature for 30 min. aCSF contained (in mM): mM: 124 NaCl, 3 KCl, 24 NaHCO3, 1.25 NaH2PO4 10 Glucose, 2.5 CaCl2, 1.3 MgSO4, saturated with 95% O2 and 5% CO2. The CA3 area was removed prior to slice transfer into the recording chamber to prevent recurrent activity.

## Electrophysiology

The recording chamber was perfused with oxygenated aCSF maintained at 32 °C. Slices were first visualised using Dodt illumination on an Olympus FV1000 BX61 microscope. Fluorescent cells were identified either in the CA1 stratum lacunosum/moleculare (neurogliaform cells) or the stratum oriens (OLM cells) using Xcite epifluorescence. Borosilicate glass pipettes (pipette resistance of 4–5 MΩ) were pulled using a horizontal P2000 (Sutter Instruments) puller. The internal solution contained (in mM) 120 KMeSO3, 8 NaCl, 10 HEPES, 4 Mg-ATP, 0.3 Na-GTP, 10 KCl,~295 mOsm, 7.4 pH. EGTA was not added to avoid introducing excess calcium buffering capacity. Internal solution aliquots were filtered on the day of experiment and small volumes of Alexa Fluor-594 (final concentration 50 µM) and Fluo-4 (final concentration 400 µM) solution aliquots were added. Recordings were obtained using a Multiclamp 700B (Molecular Devices, USA) amplifier filtered at 10 kHz and digitized at 20 kHz (National Instruments PCI-6221, USA). WinWCP 5.5.6 (John Dempster, University of Strathclyde) software was used for data acquisition. After whole-cell break-in in V-clamp, the cells were switched to I=0 mode briefly to measure the resting membrane potential, before switching to I-clamp mode for experiments. All experiments were performed in I-clamp mode, with current continuously injected to maintain cell membrane between at ~−65 mV (0 to –50 pA). All data are presented without adjustment for the junction potential (~−15 mV). Recordings were discarded if they had an access resistance >25 MΩ and if access or input resistance changed by >20% over the course of the experiment. All experiments were conducted in the presence of picrotoxin (50 µM) and CGP55845 (1 µM) to block

GABAergic transmission. D-AP5 (50 µM), tetrodotoxin (TTX; 200 nM), nimodipine (30 µM) or cyclopiazonic acid (CPA; 30 µM) were bath-applied to block NMDARs, voltage-activated sodium channels, L-type calcium channels, or to inhibit SERCA, respectively. D-AP5 was applied over a period of 7 min following a control recording, whilst TTX, nimodipine, and CPA were present in the recording chamber throughout their respective experiments.

## Two-photon imaging and uncaging

Simultaneous two-photon imaging and uncaging of MNI-caged glutamate were performed using two Ti-sapphire lasers tuned to 810 nm and 720 nm, for imaging and uncaging respectively (Mai-Tai, Spectra Physics, USA; Chameleon, Coherent, USA). MNI-caged-glutamate (3 mM; Hello Bio, UK) was added to the aCSF in a closed recirculating system upon patch-clamp break-in. 12 uncaging locations ~2 µm apart and <1 µm away from the dendrite were chosen on either side of dendrites in pseudo-random patterns. 0.5 ms-long pulses of 720 nm light were used to uncage MNI-caged-glutamate. The uncaging light pulses were separated by either 100.32ms or 0.32ms in the control asynchronous (separate responses) or the near-synchronous protocols, respectively. In the near-synchronous protocol, increasing numbers of uncaging locations were activated in a cumulative manner across sweeps 1–12. Inter-sweep interval was >30 s, during which adjustments were made to account for any focal and x-y drift. Cycles of control asynchronous and near-synchronous stimulation protocols were done one after the other to produce mean responses, without changing the order of dendritic locations uncaged. Uncaging times and locations were controlled by scanning software (Fluoview 1000 V) and a pulse generator (Berkeley Nucleonics, Ca, USA) with a Pockels cell. Calcium signals were acquired using 810 nM laser light, linescanning on the same dendritic branch as the uncaging and down-stream of the uncaging locations toward the soma. Cells were discarded upon observation of photodamage or upon change in resting membrane potential of >10 mV. All z-stacks used for the analysis of morphology and example figures were captured at the end of experiments to maximise the amount of time for experiments before cell health deterioration. Photomultiplier tube filters used: 515–560 nm (Fluo-4); 590–650 nm (Alexa-594).

## Quantification and statistical analyses

Amplitude and integral nonlinearity were regarded as the primary experimental outcomes.

Response nonlinearity was quantified using the following equation (*Cornford et al., 2019*; *Schmidt-Hieber et al., 2017*).

$$\% \, nonlinearity = \sum_{i=2}^{n} \frac{\frac{Mi}{Ai} - 1}{n - 1} * 100\%$$

where $Mi$ is the amplitude of the $i$th measured uEPSP (composed of $i$ individual uncaging spots), $Ai$ is the amplitude of the $i$th constructed arithmetically summed uEPSP, and $n$ is the total number of uncaging locations.

Seeing as the calcium signal did not return to its pre-stimulation baseline in between flashes of uncaging laser light in the control asynchronous (separate responses) condition, calculating the arithmetic sum of individual responses was not possible. Therefore, to obtain an estimate of nonlinearity for calcium traces, values from linear interpolation were used. The first sweep of the test condition, where only a single location was stimulated using uncaging light, was treated as the first value of the control asynchronous (separate responses) condition for the purposes of interpolation. The second and final value used for interpolation was from the end of the control (separate responses) trace, so that the signal would encompass all 12 uncaging events.

Calcium signals were quantified using the following equation:

$$\Delta F/A$$

where $\Delta F$ is the relative change in the Fluo-4 green channel fluorescence from baseline, and $A$ is the Alexa-594 red channel fluorescence.

uEPSPs amplitude and integral measurements were scaled to facilitate visual comparisons across dendrites. This was performed by normalising the asynchronous (separate responses) measurements from 0 to 1 using the following equation:

$$z_i = \frac{x_i - min\,(x)}{max\,(x) - min\,(x)}$$

where x is the input array, and $z_i$ is the normalised output element.

The near-synchronous responses were scaled using the same formula and the minima and maxima from the corresponding asynchronous (separate responses) conditions.

Traces including action potentials were excluded from analysis. Asynchronous separate uEPSPs were staggered by 0.82ms for arithmetic summation to replicate the near-synchronous uncaging condition. Voltage and calcium traces were filtered using a Savitzky-Golay filter. Uncaging light artefacts were removed from linescans. Due to the substantial variability in responses across dendrites, dendrites were used as the experimental unit. The numbers of dendrites, cells, and animals are reported in figure legends. $\alpha = 0.05$ was applied for all statistical tests. The t and p values are presented in the figure legends and discussed in the text, as appropriate. Estimation statistics (*Claridge-Chang and Assam, 2016*) and the generation of slopegraphs and Gardner-Altman estimation plots were performed using code provided by *Ho et al., 2019*. The slopegraphs indicate the difference in nonlinearity between asynchronous (separate responses) and the near-synchronous conditions or the differences in nonlinearity between pre- and post-drug administration. The adjacent paired mean difference plots are the group summaries. 95% confidence intervals of the mean difference were calculated by bootstrap resampling with 5000 resamples. The confidence interval is bias-corrected and accelerated. p Values are provided with estimation statistics for legacy purposes only. Data were processed using custom Python code (*Gohlke, 2025*; *Garcia et al., 2014*), Microsoft Excel, and WinWCP 5.5.6 (John Dempster, University of Strathclyde). Figures were created using custom Python code, Microsoft PowerPoint, Procreate, ImageJ.

## Acknowledgements

We are grateful to Jonathan Cornford, Vincent Magloire, Kaiyu Zheng and other members of the Department of Clinical and Experimental Epilepsy for help and advice. The authors gratefully acknowledge support from the Wellcome Trust (212285/Z/18/Z), the Medical Research Council (MR/V013556/1, MR/V034758/1) and the Gatsby Charitable Foundation (GAT3955).

## Additional information

### Funding

| Funder | Grant reference number | Author |
| --- | --- | --- |
| Wellcome Trust | 10.35802/212285 | Dimitri Michael Kullmann |
| Medical Research Council | MR/V013556/1 | Dimitri Michael Kullmann |
| Medical Research Council | MR/V034758/1 | Dimitri Michael Kullmann |
| Gatsby Charitable Foundation | GAT3955 | Dimitri Michael Kullmann |

The funders had no role in study design, data collection and interpretation, or the decision to submit the work for publication.For the purpose of Open Access, the authors have applied a CC BY public copyright license to any Author Accepted Manuscript version arising from this submission.

### Author contributions

Simonas Griesius, Conceptualization, Data curation, Investigation, Methodology, Writing – original draft, Writing – review and editing; Amy Richardson, Resources, Investigation, Methodology, Writing – review and editing; Dimitri Michael Kullmann, Conceptualization, Supervision, Funding acquisition, Writing – original draft, Writing – review and editing

### Author ORCIDs

Simonas Griesius http://orcid.org/0000-0002-2025-1064
Amy Richardson https://orcid.org/0000-0002-1552-2915

Dimitri Michael Kullmann  https://orcid.org/0000-0001-6696-3545

### Ethics

Research was conducted in accordance with the Animal (Scientific Procedures) Act 1986 Amendment Regulations (SI 2012/3039) under PPL: PP3944290.

Reviewer #2 (Public review): https://doi.org/10.7554/eLife.100268.3.sa1
Reviewer #3 (Public review): https://doi.org/10.7554/eLife.100268.3.sa2
Author response https://doi.org/10.7554/eLife.100268.3.sa3

## Additional files

### Supplementary files

MDAR checklist

### Data availability

Electrophysiology and calcium fluorescence data for Figures 1-8 together with uncaging distances, dendrite orders and dendrite diameters, are available at https://doi.org/10.5522/04/28143356.v2 under the Creative Commons CC0 licence.

The following dataset was generated:

| Author(s) | Year | Dataset title | Dataset URL | Database and Identifier |
|-----------|------|---------------|-------------|-------------------------|
| Griesius S, Richardson A, Kullmann DM | 2025 | Supralinear dendritic integration in murine dendrite-targeting interneurons: voltage and calcium fluorescence nonlinearities in response to near-synchronous 2-photon glutamate uncaging | https://doi.org/10.5522/04/28143356.v2 | UCL Research Data Repository, 10.5522/04/28143356.v2 |

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
