## [Editor Report · eLife Assessment]

In this manuscript, Griesius et al analyze the dendritic integration properties of NDNF and OLM interneurons, and suggest that the supralinear NMDA receptor-dependent synaptic integration may be associated with dendritic calcium transients only in NDNF interneurons. These findings are **important** because they suggest there might be functional heterogeneities in the mechanisms underlying synaptic integration in different classes of interneurons of the mouse neocortex and hippocampus. The revised work remains **incomplete** due to remaining concerns about experimental methodology, cell health, and lack of dendritic Na-spikes which have been recorded in previous works.

---

## [Referee Report · Reviewer #2 (Public review)]

Summary:

Griesius et al. investigate the dendritic integration properties of two types of inhibitory interneurons in the hippocampus: those that express NDNF+ and those that express somatostatin. They found that both neurons showed supralinear synaptic integration in the dendrites, blocked by NMDA receptor blockers but not by blockers of Na+ channels. These experiments are critically overdue and very important because knowing how inhibitory neurons are engaged by excitatory synaptic input has important implications for all theories involving these inhibitory neurons.

Comments on revisions:

The authors have addressed the reviewers' comments, but haven't resolved most of the key issues.

Specifically, performing only a single uncaging experiment at a single dendritic location per cell prevents a detailed biophysical analysis of NDNF and OLM cell integration properties. A more extended exploration would have potentially addressed several of the reviewers' questions. It is particularly worrying that the authors cite cell health, dendritic blebbing, and changes in input resistance as the reason for terminating experiments after a single uncaging event. This suggests that the uncaging laser may be damaging the dendrite, potentially affecting the membrane potential directly, and overall cell health, beyond simply uncaging glutamate.

While the authors' qualitative conclusions about supra-linear integration and NMDA receptor dependency seem plausible, the limited data and potential methodological issues weaken any quantitative interpretations and comparisons between the two cell types.

Similarly, the absence of dendritic Na-spikes remains unexplained, despite reports of strong dendritic Na-currents in these cells.

---

## [Referee Report · Reviewer #3 (Public review)]

Summary:

The authors study temporal summation of caged EPSPs in dendrite-targeting hippocampal CA1 interneurons. The data indicate non-linear summation, which is larger in dendrites of NDNF-expressing neurogliaform cells versus OLM cells. However, the underlying mechanisms are largely unclear.

Strengths:

Synaptic integration in dendrites of cortical GABAergic interneurons is important and still poorly investigated. Focal 2-photon uncaging of glutamate is a nice and detailed method to study temporal summation of small potentials in dendritic segments. 2P calcium imaging is a powerful method to potentially disentangle dendritic signal processing in interneuron dendrites.

Weaknesses:

Due to several experimental limitations of the study including a relatively low number of recorded dendrites, lack of voltage-clamp recordings, lack of NMDA-dependent calcium signals in OLM cells and lack of wash-out during pharmacological experiments (AP5-application), the mechanistic insights are limited.

(1) NMDA-receptor signalling in NDNF-IN. The authors nicely show that temporal summation in dendrites of NDNF-INs is to a certain extent non-linear. Pharmacology with AP5 hints towards contribution of NMDA receptors. However, the authors report that the non-linearity in not significantly dependent on EPSP amplitude (Fig. S2), which should be the case if NMDA-receptors are involved. Unfortunately, there are no voltage-clamp data showing NMDA and AMPA currents, potentially providing a mechanistic explanation for the non-linear summation.

(2) Recovery of drug effect. Pharmacological application of AP5 is the only argument for the involvement of NMDA receptors. However, as long-lasting experiments were apparently difficult to obtain, there is no washout-data presented - only drug effect versus baseline. For all the other drugs (TTX, Nimodipine, CPA) recordings were even shorter, lacking a baseline recording. Thus, it remains open to what extent the AP5-effect might be affected by rundown of receptors or channels during whole-cell recordings or beginning phototoxicity.

(3) Nonlinear EPSP summation in OLM-IN. The authors do similar experiments in dendrite-targeting OLM-INs and show that the non-linear summation is smaller than in NDNF cells. The reason for this remains unclear. The diameter of proximal dendrites in OLM cells is larger than the diameter in NDNF cells. However, there is probably also an important role of synapse density and glutamate receptor density, which was shown to be very low in proximal dendrites of OLM cells and strongly increase with distance (Guirado et al. 2014, Cerebral Cortex 24:3014-24, Gramuntell et al. 2021, Front Aging Neurosci 13:782737). Therefore, it would have been helpful to see experiments quantifying synapse density (counting spines, PSD95-puncta, ...) and show how this density compares with non-linearity in the analyzed NDNF and OLM dendrites.

(4) NMDA in OLM-IN. Similar to the NDNF cells, the authors argue for an involvement of NMDA receptors in OLM cells, based on bath-application of AP5 (Fig. 8). Again, there seems to be no significant dependence on EPSP amplitude (Fig. S3). Even more remarkable, the authors claim that there is no dendritic calcium increase after activation of NMDA receptors without showing data. Therefore, it remains unclear whether the calcium signals are just below detection threshold, or whether the non-linearity depends on other calcium-impermeable channels and receptors. To understand this phenomenon different calcium sensors, different Ca2+/Mg2+ concentrations or voltage-clamp data would have helped.

---

## [Author Response]

The following is the authors’ response to the original reviews.

**Public Reviews:**

**Reviewer #1 (Public Review):**
The manuscript by Griesius et al. addresses the dendritic integration of synaptic input in cortical GABAergic interneurons (INs). Dendritic properties, passive and active, of principal cells have been extensively characterized, but much less is known about the dendrites of INs. The limited information is particularly relevant in view of the high morphological and physiological diversity of IN types. The few studies that investigated IN dendrites focused on parvalbumin-expressing INs. In fact, in a previous study, the authors examined dendritic properties of PV INs, and found supralinear dendritic integration in basal, but not in apical dendrites (Cornford et al., 2019 eLife).In the present study, complementary to the prior work, the authors investigate whether dendrite-targeting IN types, NDNF-expressing neurogliaform cells, and somatostatin(SOM)-expressing O-LM neurons, display similar active integrative properties by combining clustered glutamate-uncaging and pharmacological manipulations with electrophysiological recording and calcium imaging from genetically identified IN types in mouse acute hippocampal slices.The main findings are that NDNF IN dendrites show strong supralinear summation of spatially- and temporally-clustered EPSPs, which is changed into sublinear behavior by bath application of NMDA receptor antagonists, but not by Na+-channel blockers. L-type calcium channel blockers abolished the supralinear behavior associated calcium transients but had no or only weak effect on EPSP summation. SOM IN dendrites showed similar, albeit weaker NMDA-dependent supralinear summation, but no supralinear calcium transients were detected in these INs. In summary, the study demonstrates that different IN types are endowed with active dendritic integrative mechanisms, but show qualitative and quantitative divergence in these mechanisms.While the research is conceptionally not novel, it constitutes an important incremental gain in our understanding of the functional diversity of GABAergic INs. In view of the central roles of IN types in network dynamics and information processing in the cortex, results and conclusions are of interest to the broader neuroscience community.The experiments are well designed, and closely follow the approach from the previous publication in parts, enabling direct comparison of the results obtained from the different IN types. The data is convincing and the conclusions are well-supported, and the manuscript is very well-written.I see only a few open questions and some inconsistencies in the presentation of the data in the figures (see details below).

We thank the reviewer for the evaluation and address the detailed points below.

**Reviewer #2 (Public review):**
Summary:Griesius et al. investigate the dendritic integration properties of two types of inhibitory interneurons in the hippocampus: those that express NDNF+ and those that express somatostatin. They found that both neurons showed supralinear synaptic integration in the dendrites, blocked by NMDA receptor blockers but not by blockers of Na+ channels. These experiments are critically overdue and very important because knowing how inhibitory neurons are engaged by excitatory synaptic input has important implications for all theories involving these inhibitory neurons.Strengths:(1) Determined the dendritic integration properties of two fundamental types of inhibitory interneurons.(2) Convincing demonstration that supra-threshold integration in both cell types depends on NMDA receptors but not on Na+ channels.Weaknesses:It is unknown whether highly clustered synaptic input, as used in this study (and several previous studies), occurs physiologically.

We are grateful to the reviewer for the critique. Indeed, the degree to which clustered inputs belonging to a functional neuronal assembly occur on interneuron dendrites is an open question. However, Chen et al (2013, Nature 499:295-300) reported that dendritic domains of PV-positive interneurons in visual cortex, unlike their somata, exhibit calcium transients in vivo which are highly tuned to stimulus orientation. This suggests that clustered inputs to dendritic segments may well belong to functional assemblies, much as in principal cells (e.g. Wilson et al, 2016, Nature Neuroscience 19:1003–1009; Iacaruso et al, 2017, Nature 547;449–452). In our earlier work reporting NMDAR-dependent supralinear summation of glutamate uncaging-evoked responses at a subset of dendrites on PV-positive interneurons, we demonstrated how this arrangement in an oscillating feedback circuit could be exploited to stabilise neuronal assemblies.

**Reviewer #3 (Public review):**
Summary:The authors study the temporal summation of caged EPSPs in dendrite-targeting hippocampal CA1 interneurons. There are some descriptive data presented, indicating non-linear summation, which seems to be larger in dendrites of NDNF expressing neurogliaform cells versus OLM cells. However, the underlying mechanisms are largely unclear.Strengths:Focal 2-photon uncaging of glutamate is a nice and detailed method to study temporal summation of small potentials in dendritic segments.Weaknesses:(1) NMDA-receptor signaling in NDNF-IN. The authors nicely show that temporal summation in dendrites of NDNF-INs is to a certain extent non-linear. However, this non-linearity varies massively from cell to cell (or dendrite to dendrite) from 0% up to 400% (Figure S2). The reason for this variability is totally unclear. Pharmacology with AP5 hints towards a contribution of NMDA receptors. However, the authors claim that the non-linearity is not dependent on EPSP amplitude (Figure S2), which should be the case if NMDA-receptors are involved. Unfortunately, there are no voltage-clamp data of NMDA currents similar to the previous study. This would help to see whether NMDA-receptor contribution varies from synapse to synapse to generate the observed variability? Furthermore, the NMDA- and AMPA-currents would help to compare NDNF with the previously characterized PV cells and would help to contribute to our understanding of interneuron function.

We thank the reviewer for the helpful comments.

We did not actually claim that EPSP amplitude has no role in determining the magnitude of non-linearity: “Among possible sources of variability for voltage supralinearity, we did not observe a systematic dependence on the average amplitude of individual uEPSPs […] (Fig. S2)”. Whilst we fully agree that, at first sight, a positive dependence of supralinearity on uEPSP amplitude might be expected simply from the voltage-dependent kinetics of NMDARs, there are two main reasons why this could have been obscured. First, the expected relationship is non-monotonic, because with large local depolarizations the driving force collapses, as seen in the overall sigmoid shape of the average relationship between the scaled observed response and arithmetic sum (e.g. Figs 2a & c; 4c & e). Therefore, we would arguably expect a parabolic relationship rather than a simple positive slope relating the degree of supralinearity to the average amplitude of individual uEPSPs. Second, given that the uncaging distance varied substantially, the average amplitudes of the individual uEPSPs recorded at the soma would have undergone different degrees of electrotonic attenuation and further distortion by active conductances before they were measured. Ultimately, the plots in Fig. S2 show too much scatter to be able to exclude a positive or parabolic relationship of nonlinearity to uEPSP amplitude. To avoid misunderstanding, we have changed the sentence in the Results that refers to Fig. S2 to: “Among possible sources of variability for voltage supralinearity, we did not observe a significant monotonic dependence on the average amplitude of individual uEPSPs, distance from the uncaging location along the dendrite to the soma, [or] the dendrite order (Fig. S2)”.

As for the relative contributions of NMDARs and AMPARs, voltage clamp recordings from both neurogliaform and OLM interneurons have already been reported, with the conclusion that neurogliaform cells exhibit relatively larger NMDAR-mediated currents (e.g. Chittajallu et al. 2017; Booker et al. 2021; Mercier et al. 2022), entirely in keeping with the conclusions of our study. Repeating these measurements would add little to the study. Furthermore, because the mean baseline uEPSP amplitude was <0.5 mV (Fig S2), it would be difficult to obtain reliable meaurements of isolated NMDAR-mediated uEPSCs.

Turning to the high variability of supralinearity, indeed, the 95% confidence interval for the data in Fig. 2d is 73%, 213%. This degree of variability is consistent with the wide range of NMDAR/AMPAR ratios reported by Chittajallu et al. 2017 (their Fig. 1g), compounded by the expected non-monotonic relationship alluded to above.

(2) Sublinear summation in NDNF-INs. In the presence of AP5, the temporal summation of caged EPSPs is sublinear. That is potentially interesting. The authors claim that this might be dependent on the diameter of dendrites. Many voltage-gated channels can mediate such things as well. To conclude the contribution of dendritic diameter, it would be helpful to at least plot the extent of sublinearity in single NDNF dendrites versus the dendritic diameter. Otherwise, this statement should be deleted.

We have plotted the degree of nonlinearity against dendritic diameter for neurogliaform cells (under baseline conditions and in D-AP5) in Fig S2h-k. We did not observe any significant linear correlations, other than between amplitude nonlinearity and dendrite diameter post D-AP5. This does not negate the possibility that the significant difference in average dendritic diameters between neurogliaform and OLM cells contributes to differences in impedance (which we have rephrased as “Among possible explanations is that the local dendritic impedance is greater in neurogliaform cells, lowering the threshold for recruitment of regenerative currents”).

(3) Nonlinear EPSP summation in OLM-IN. The authors do similar experiments in dendrite-targeting OLM-INs and show that the non-linear summation is smaller than in NDNF cells. The reason for this remains unclear. The authors claim that this is due to the larger dendritic diameter in OLM cells. However, there is no analysis. The minimum would be to correlate non-linearity with dendritic diameter in OLM-cells. Very likely there is an important role of synapse density and glutamate receptor density, which was shown to be very low in proximal dendrites of OLM cells and strongly increase with distance (Guirado et al. 2014, Cerebral Cortex 24:3014-24, Gramuntell et al. 2021, Front Aging Neurosci 13:782737). Therefore, the authors should perform a set of experiments in more distal dendrites of OLM cells with diameters similar to the diameters of the NDNF cells. Even better would be if the authors would quantify synapse density by counting spines and show how this density compares with non-linearity in the analyzed NDNF and OLM dendrites.

The difference in average dendritic diameters between OLM and neurogliaform cells is highly significant (Fig. 8q, P<0.001). We do not claim that dendritic diameter (and by implication local impedance) is the only determinant of the degree of non-linearity. The suggestion that a gradient of glutamate receptor density contributes is interesting. However, the results of uncaging experiments targeting more distal OLM dendrites of similar diameter as neurogliaform dendrites would be subject to numerous confounds, not least the very different electrotonic attenuation, likely differences in various active conductances, and the presence of spines in OLM dendrites (which are generally sparse and were not reliably imaged in our experiments). Moreover, the cell would have to remain patched for longer in order for the fluorescent dyes to invade the distal dendrites. This alone could potentially result in systematic biases among groups. We now cite Guirrado et al (2014) and Gramuntell et al (2021) to highlight that factors other than dendritic diameter per se, such as inhomogeneity in spine and NMDA receptor density may also contribute to the heterogeneity of nonlinear summation in OLM cells.

(4) NMDA in OLM. Similar to the NDNF cells, the authors claim the involvement of NMDA receptors in OLM cells. Again there seems to be no dependence on EPSP amplitude, which is not understandable at this point (Figure S3). Even more remarkable is the fact that the authors claim that there is no dendritic calcium increase after activation of NMDA receptors. Similar to NDNF-cell analysis there are no NMDA currents in OLMs. Unfortunately, even no calcium imaging experiments were shown. Why? Are there calcium-impermeable NNDA receptors in OLM cells? To understand this phenomenon the minimum is to show some physiological signature of NMDA-receptors, for example, voltage-clamp currents. Furthermore, it would be helpful to systematically vary stimulus intensity to see some calcium signals with larger stimulation. In case there is still no calcium signal, it would be helpful to measure reversal potentials with different ion compositions to characterize the potentially 'Ca2+ impermeable' voltage-dependent NMDA receptors in OLM cells.

The same response to point (1) above applies to OLM cells. As with neurogliaform cells, mean OLM baseline asynchronous (separate response) amplitudes were <0.5 mV, making it very difficult to record an isolated NMDAR-mediated uEPSC. Having said that, NMDARs do contribute to EPSCs elicited by stimulation of multiple afferents (e.g. Booker et al, 2021). We do not claim that dendritic calcium transients cannot be elicited following activation of NMDARs in OLM cells. We simply reported that the evoked uEPSPs, designed to approximate individual synaptic signals, were sub-threshold for detectable dendritic calcium signals under conditions that were suprathreshold in neurogliaform cells. The statement has been amended to specify that there were no detectable signals under our recording conditions. There is no evidence presented in the manuscript to suggest that OLM NMDARs are calcium impermeable and indeed no such claim was made.

**Recommendations for the authors:**

**Reviewer #1 (Recommendations for the authors):**
There is a large variability in the observed dendritic nonlinearity, in NDNF IN dendrites e.g. the uEPSP amplitude nonlinearity measure varies from as low as 10-20% to over 200%. As only single dendrites were recorded from each IN, it is unclear if this variability is among the cells or between individual dendrites. While the authors analyzed some potential factors, such as distance along the dendrites, branch order, or response magnitude (amplitude and integral), they did not find any substantial correlation. It remains open if different dendrites of NDNF INs, located in the str. moleculare vs. those in or projecting towards str. radiatum, have divergent properties. Similarly, for SOM INs an important question is if axon-carrying dendrites show distinct properties.In this context, it would be interesting to see not only values for the mean nonlinearity but also the maximal nonlinearity and its distribution.

Nonlinearity as defined in the manuscript is a cumulative measurement. The final value per dendritic segment is therefore the sum of nonlinearities at 1 to 12 near-synchronous uncaging locations. The data for the individual dendritic segments are shown in the slopegraphs as in Fig 2b, with their distribution visible. The averages referred to in the results correspond to the paired mean difference plots, which are the group summaries. The method section has been amended to clarify the analysis method. We did not address specifically whether dendrites projecting in different directions behaved differently. This is an interesting question beyond the scope of this study. Nor did we compare axon-carrying OLM dendrites to other dendrites.

Figures:Figure 1: The gray line in plots g and h is not explained. While it looks like an identity line, the legend in plot i ("asynchronous") interferes.

In plots g and h the gray line is the line of identity. In plot i it is an estimate of the linear summation. In plot i it is not the line of identity as it does not start at the origin with a slope of 1. The figure legend has been amended to clarify.

In the same panels (Figure 1g,h, and subsequent figures) consider changing the title from "soma (voltage)" to uEPSP.

The titles have been amended.

In panel Figure 1i note the missing " in the title.

Title amended.

In panel Figure 1h: Shouldn't the X-axis label and legend text read "Arithmetic sum of (EPSP) integrals" instead of "Integral of arithmetic sum".

The wording more accurately reflects the analytical operations. The asynchronous (separate) responses were summed arithmetically first, and then the integral was taken of each cumulative sum. We have therefore left the axis title and legend unchanged.

Figure 2a,c: Could you please describe how the scaling was performed for the two axes?

Method section amended.

In the same panels (Figure 2a,c, and subsequent figures), the legend seems to be misleading: the plot is NS Amplitude/Integral vs Arithmetic sum, and the black line is the identity line (or scaled interpolation of the arithmetic sum, which is essentially the same).

The scaled arithmetic sums (uEPSP amplitude, integral) represent linear summation and so overlap with the line of identity. The interpolation estimate of the asynchronous (separate) calcium transient response does not overlap with the line of identity as this estimate does not start at the origin with a slope of 1. The legends throughout the manuscript have been amended to clarify this.

Figure 2b,d,f (and subsequent figures) slope plots: Please indicate that this is the average amplitude supralinearity for the individual recorded dendrites. Note here that the Results text mentions only the average amplitude supralinearity, but not the slop plots, paired mean difference, or Gardner-Altman estimation, illustrated in the figures.

Nonlinearity as defined in the manuscript is a cumulative measurement. The final value per dendritic segment is therefore the sum of nonlinearities at 1 to 12 near-synchronous uncaging locations. The data for the individual dendritic segments are shown in the slopegraphs as in fig 2b, with their distribution visible. The averages referred to in the results correspond to the paired mean difference plots, which are the group summaries. The method section has been amended to clarify.

Fig 2e: The legend (both text and figure, also in the following figures) is confusing, as the gray line and diamonds are defined as separate 12(?) responses, but it seems to represent a linear interpolation of the scaled arithmetic sums (ultimately nothing else but an identity line).

The grey line shows the linear interpolation output between the calcium transient measurements at 1 uncaging location and at 12 uncaging locations. The 12th uncaging location is indicated in the key as “separate 12”. The linear interpolation in these plots does represent linear summation but is not the line of identity as it does not begin at the origin and does not have a slope of 1.

**Reviewer #2 (Recommendations for the authors):**
This study is well-developed and technically executed. I only have minor comments for the authors:(1) To target NDNF+ neurons, the authors use the NDNF-Cre mouse line and a Cre-dependent AAV using the mDLX promotor. Why the mDLX promotor? Would it have been sufficient to use any Cre-dependent fluorophore?

Pilot experiments revealed leaky expression when a virus driving flexed ChR2 under a non-specific promoter (EF1a) was injected in the neocortex of Ndnf-Cre mice (Author response image 1). In our hands, and in line with Dimidschtein et al (2016), the use of the mDLX enhancer reduced off-target expression.

**Author response image 1. sa3fig1:** A. AAV2/5-EF1a-DIO-hChR2(H134R)-mCherry injected into superficial neocortex of Ndnf-Cre mice led to expression in a few pyramidal neurons in addition to layer 1 neurogliaform cells. B. Patch-clamp recording from a non-labelled pyramidal cell showed that an optogenetically evoked glutamatergic current remained after blockade of GABAA and GABAB receptors, further confirming limited specificity of expression of ChR2. (Data from M Muller, M Mercier and V Magloire, Kullmann lab.)

(2) The distance of the uncaring sites from the soma plays a key role. The authors should indicate the mean distance of the cluster and its variance.

Uncaging distance from soma is indicated for both NGF and OLM interneurons in the supplementary figures S2 and S3 respectively.

(3) Martina et al., in Science 2000, showed high levels of Na+ channels in the dendrites of OLM cells and hinted that spikes could occur in them. The authors should discuss this possible discrepancy.

Discussion amended.

(4) Looking at Figure 1d, the EPSPs look exceptionally long-lasting, longer than those observed by stimulating axonal inputs. Could this indicate spill-over excitation? If so, how could this affect the outcome of this study?

The asynchronous (separate responses) decay to baseline within 100 ms, similar to the neurogliaform EPSPs evoked by electrical stimulation of axons in the SLM in Mercier et al. 2022. We observed clear plateau potentials in a minority of cells (e.g. Fig. S1b). Such plateau potentials can be generated by dendritic calcium channels and we do not consider that glutamate spillover needs to be invoked to account for them.

(5) In the legend of Figure 2: "n=11 dendrites in 11 cells from 9 animals". Why do the authors only study 11 dendrites from 11 cells? Isn't it possible to repeatedly stimulate clusters of synaptic inputs onto the same cells? In principle, could one test many dendrites of the same cell at different distances from the soma? It is also remarkable that there were very few cells per animal.

The goal always was to record from as many dendrites as possible from the same cells whilst maintaining high standards of cell health. When cell health indicators such as blebbing, input resistance change or resting voltage change were detected, no further dendritic location could be tested with reasonable confidence. In a given 400 um slice there would be relatively few healthy candidate cells at a suitable depth to attempt to patch-clamp.